# CD1d-dependent rewiring of lipid metabolism in macrophages regulates innate immune responses

Phillip M. Brailey[1,2,4], Lauren Evans[1,2,4], Juan Carlos López-Rodríguez[1,2], Anthony Sinadinos[1,2], Victoria Tyrrel[3], Gavin Kelly [2], Valerie O'Donnell [3], Peter Ghazal [3], Susan John [1] & Patricia Barral [1,2] ✉

Alterations in cellular metabolism underpin macrophage activation, yet little is known regarding how key immunological molecules regulate metabolic programs in macrophages. Here we uncover a function for the antigen presenting molecule CD1d in the control of lipid metabolism. We show that CD1d-deficient macrophages exhibit a metabolic reprogramming, with a down-regulation of lipid metabolic pathways and an increase in exogenous lipid import. This metabolic rewiring primes macrophages for enhanced responses to innate signals, as CD1d-KO cells show higher signalling and cytokine secretion upon Toll-like receptor stimulation. Mechanistically, CD1d modulates lipid import by controlling the internalization of the lipid transporter CD36, while blocking lipid uptake through CD36 restores metabolic and immune responses in macrophages. Thus, our data reveal CD1d as a key regulator of an inflammatory-metabolic circuit in macrophages, independent of its function in the control of T cell responses.

The metabolic and immune pathways are strongly interconnected and cross-regulated in macrophages. While lipids have classically been regarded as exogenous elements in immune responses, in recent years it has become clear that lipid metabolism and the cellular lipid status play a central role in the regulation of macrophage activation and function[1]. Lipids function as an energy source as well as components for cellular membranes, but they also regulate signal transduction and gene expression in macrophages. Consequently, dysregulated lipid metabolism in macrophages is associated with a variety of pathologies including obesity, atherosclerosis, cancer or infections.

Macrophages rapidly and profoundly reprogramme their lipid metabolism in response to activation signals. Accordingly, activation of macrophages by pattern recognition receptors (such as Toll-like receptors (TLR)) or cytokines rapidly induces changes to their lipid metabolic and energetic programmes, and this reprogramming of lipid homoeostasis is critical to control macrophage functions[1–5]. For instance, viral infection, TLR ligation or type I interferon (IFN) stimulation have been shown to rewire macrophages' lipid metabolism by downregulating de novo fatty acid and cholesterol synthesis and upregulating lipid import[3,4,6]. Also, IL-4 stimulation induces de novo lipid synthesis, which is a necessary step for macrophage alternative activation[7]. Conversely, genetic or pharmacological interference with this lipid metabolic reprograming controls macrophages' ability to mediate inflammatory responses. As such, genetic manipulation of the cholesterol biosynthetic pathway in macrophages alters the balance between lipid synthesis and import, and leads to increased cytokine secretion in response to TLR ligation, spontaneous IFN-β production and increased protection from viral challenge[3,4]. On the other hand, lipid metabolic pathways are also essential for the induction of trained immunity in myeloid cells, and manipulation of lipid pathways prevents trained immunity induction[8,9]. Thus, it is clear that a crosstalk between immunity and lipid metabolism functions as a critical component in host defence and inflammation. Nonetheless, our knowledge of this immuno-metabolic circuit is still limited and the molecular pathways controlling this crosstalk and how they regulate immune cell function remain poorly understood.

[1]The Peter Gorer Department of Immunobiology, King's College London, London, UK. [2]The Francis Crick Institute, London, UK. [3]School of Medicine, Cardiff University, Cardiff, UK. [4]These authors contributed equally: Phillip M. Brailey, Lauren Evans. ✉e-mail: patricia.barral@kcl.ac.uk

Structurally related to MHC class I, the antigen presenting molecule CD1d binds and presents lipid antigens to Natural Killer T (NKT) cells[10]. CD1d is constitutively expressed in hematopoietic (macrophages, B cells, dendritic cells) as well as on non-hematopoietic cells (such as intestinal epithelial cells (IECs)). While the role of CD1d in lipid presentation to NKT cells is well-established, growing evidence supports additional "non-classical" roles for CD1d which impact the functions of CD1d-expressing cells[11–14]. For example, ligation of CD1d in IECs leads to STAT-3-dependent secretion of IL-10 rendering protective effects in murine models of inflammatory bowel disease[15]. Interestingly, alterations in the levels of expression and/or intracellular trafficking of CD1d have been observed in autoimmune diseases as well as during viral and bacterial infections[16,17], suggesting that the strict control of CD1d expression and function are key components in host defence and inflammatory responses.

In this study we uncovered a function for CD1d as a molecular link between innate immunity and lipid metabolism. We have found that CD1d deficiency in primary macrophages leads to increased secretion of cytokines and increased activation of MAP-Kinase and NF-kB pathways in response to TLR stimulation. Unexpectedly, the hyperresponsiveness of CD1d-KO cells is underpinned by dysregulation of lipid metabolic pathways. CD1d deficiency reprogrammes lipid metabolism, downregulating lipid pathways and the expression of the transcription factor peroxisome proliferator-activated receptor delta (PPARδ). Conversely, CD1d-deficient macrophages show an increase in lipid uptake mediated by the transporter CD36, while inhibition of CD36 restores both metabolic and immune responses. Thus, our data reveal a function for CD1d in controlling lipid metabolism and inflammatory responses in macrophages.

## Results

### CD1d contributes to the negative regulation of TLR responses in myeloid cells

In addition to the well-established functions of CD1d molecules in lipid presentation to T cells, recent literature supports *unconventional* roles for CD1d, which can regulate the activation and function of CD1d-expressing cells[11–13,15]. Accordingly, recent publications proposed that CD1d may regulate inflammatory responses in myeloid cells[14,18], however the data in the literature are conflicting and the outcome and underpinning mechanisms of the CD1d-dependent control of macrophage functions remain unresolved. To investigate the role of CD1d in the regulation of TLR responses, we took advantage of primary macrophages isolated from the peritoneal cavity (pMacs) of CD1d-deficient (CD1d-KO) mice or littermate controls. Flow cytometry characterization of WT and CD1d-KO pMacs showed comparable expression of various surface markers (e.g. MHC-I, CD24, CD86, CD80) as well as TLR4 (Fig. 1a, Supplementary Fig 1a, b). To evaluate the effect of TLR stimulation in WT and CD1d-KO cells, we purified pMacs, cultured them with the TLR4 agonist LPS, and measured cytokine secretion and mRNA expression 6 h after stimulation (Fig. 1b, Supplementary Fig 1c). To get a broad overview of the cytokine profile secreted by WT and CD1d-KO pMacs in response to TLR stimulation we took advantage of a cytometric bead array which enables simultaneous quantification of 13 different cytokines/chemokines (IL-6, TNF-α, MCP-1, IL-1α, IL-1β, IL-10, IL-12p70, IL-17A, IL-23, IL-27, GM-CSF, IFN-γ, IFN-β). Strikingly, CD1d-KO pMacs showed an increased secretion of the type I interferon IFN-β (~2-fold) and of the proinflammatory cytokines IL-6 and TNF-α (~1.5-fold) in comparison to WT cells, while secretion of IL-10 was comparable in WT vs. KO pMacs (Fig. 1b, Supplementary Fig 1c; note that all other cytokines were undetectable in our essays). Consistent with these results, CD1d-KO cells also showed increased expression of the interferon stimulated genes (ISG) *Ifit1* and *Mx1* as detected by qPCR. Thus, these data suggest a role for CD1d in the regulation of TLR responses in macrophages.

CD1d-KO mice lack CD1d expression in all cells -as well as NKT cells- which could directly or indirectly control the development and/or function of other immune cells. Thus, to explore whether the differences in TLR responses between CD1d-deficient and sufficient macrophages are cell-intrinsic, we generated mixed bone marrow (BM) chimeras (WT:CD1d-KO; 50:50) by co-transferring WT (CD45.1+) and CD1d-KO (CD45.2+) BM into irradiated recipients (CD45.1+CD45.2+). WT and CD1d-KO pMacs were sort-purified from chimeras on the basis of their congenic marker expression and stimulated with LPS (Fig. 1c, Supplementary Fig 1d). In keeping with our previous results, CD1d-KO cells isolated from chimeric mice showed increased secretion of IL-6 as well as increased expression of ISGs and *Il6* mRNA in comparison to WT cells, supporting a cell-intrinsic role for CD1d in TLR regulation.

To further confirm the function of CD1d in TLR responses, we took advantage of GM-CSF cultured bone-marrow derived cells (BMDCs) generated from WT and CD1d-KO mice (Fig. 1d, Supplementary Fig 2). Consistent with the results obtained for pMacs, after stimulation with LPS, CD1d-KO BMDCs secreted more type I IFN and pro-inflammatory cytokines (IL-6, TNF-α, MCP-1) and showed increased expression of ISGs as well as *Il6* and *Tnfa* mRNA than WT BMDCs (Fig. 1d, Supplementary Fig 2b–e). This increase in cytokine secretion was independent of TLR4 expression, as its levels were comparable in WT and CD1d-KO BMDCs by both flow-cytometry (Supplementary Fig 2a) and qPCR (Supplementary Fig 2f). The augmented secretion of IFN-β by CD1d-KO BMDCs was also detected upon stimulation with the TLR9 agonist CpG, while the TLR3 agonist poly(I:C) or alive *E. coli*, induced significantly increased cytokine secretion (IL-6, TNF-α), suggesting a broad effect for CD1d in the modulation of TLR responses (Fig. 1d, Supplementary Fig 2b–e). Moreover, the hyperactivation of CD1d-KO cells in response to TLR stimulation was obvious either when using littermate mice as controls, or age and sex matched WT C57BL/6 mice (Supplementary Fig 2g). On the other hand, stimulation of BMDCs with cytokines (IFN-β, TNF-α or IFN-γ) resulted in comparable expression of ISGs and cytokines in WT and CD1d-KO cells (Supplementary Fig 2h). To validate the function of CD1d in the regulation of TLR-mediated responses, we repeated these experiments with BMDCs obtained from an independently generated CD1d-deficient mouse line[19,20] bred in a different animal facility (Supplementary Fig 2i). As observed in our previous experiments, CD1d-deficient BMDCs from this mouse line also secreted more cytokines than cells from littermate control animals when stimulated with TLR ligands, confirming the specificity of this phenotype.

Finally, we explored the function of CD1d in inflammatory responses in vivo, by using a model of LPS-induced inflammation. Since NKT cells participate in the early cytokine response to TLR ligands[21,22] we utilised an approach which overcomes the lack of NKT cells in CD1d-KO mice (Fig. 1e). We depleted WT mice of endogenous macrophages via pre-treatment with clodronate liposomes and replenished them by adoptively transferring CD1d-sufficient or deficient BMDCs[23]. After LPS challenge, mice reconstituted with CD1d-KO cells showed increased susceptibility to LPS-induced inflammation evidenced by decreased body temperature and higher concentrations of cytokines in their blood compared to mice reconstituted with WT cells (Fig. 1e).

Thus, collectively our data support a cell-intrinsic role for CD1d in the negative regulation of TLR responses in myeloid cells.

### Enhanced signalling in CD1d-deficient cells in response to TLR stimulation

Next, we investigated whether CD1d interferes with TLR signalling pathways in macrophages. To test this, we examined the activation of the mitogen-activated protein kinase (MAPK) and NF-κB pathways, which are both downstream of TLR signalling. Following LPS stimulation, activation of the MAPKs p38 and ERK and of the NF-κB subunit p65 was enhanced in CD1d-KO pMacs in comparison to WT cells as evidenced by an increase in phosphorylation of these molecules (Fig. 2a). We obtained similar results in time-course experiments

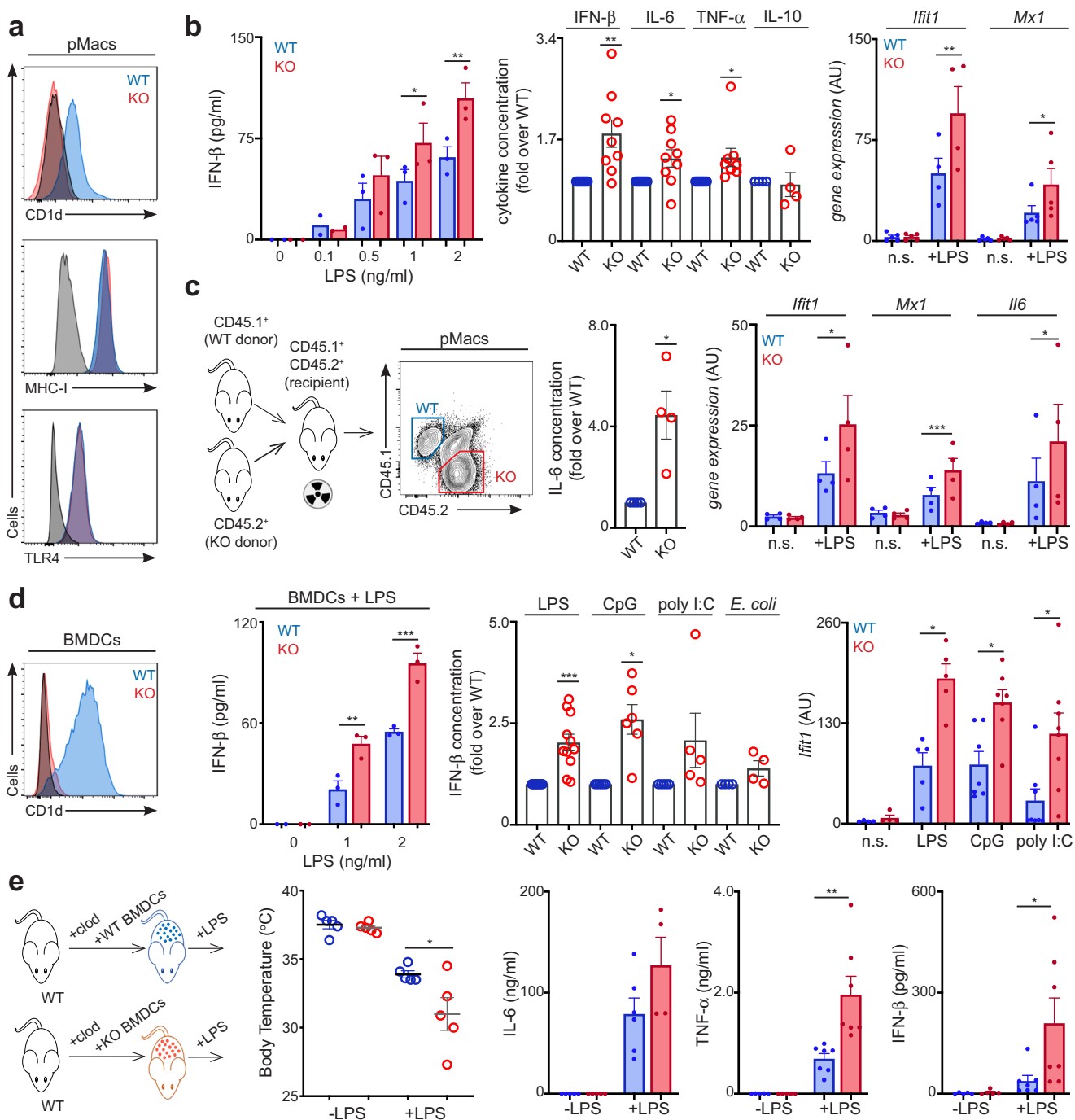

**Fig. 1 | A cell-intrinsic function for CD1d in the regulation of TLR responses.**
**a**, **b** Peritoneal macrophages (pMacs) were obtained from WT and CD1d-KO mice. **a** Representative flow cytometry of WT (blue) and CD1d-KO (red) pMacs showing expression of the depicted markers in steady state. **b** Secretion of IFN-β (left, $n = 2-3$), normalised concentration of secreted cytokines (fold over WT, middle, $n = 4-9$) and gene expression (right, $n = 4-5$) in WT and CD1d-KO pMacs cultured with LPS for 6 h. *$p < 0.05$; **$p < 0.01$, 2-way ANOVA (left), one-sample (middle) or two-tailed paired *t*-test (right). n.s. = non-stimulated. **c** Mixed BM chimeras (WT:CD1d-KO; 50:50) were generated by co-transferring WT (CD45.1⁺) and CD1d-KO (CD45.2⁺) BM into irradiated recipients (CD45.1⁺CD45.2⁺). WT and KO pMacs were sort-purified from chimeric mice and stimulated with LPS. Normalised concentration of IL-6 (fold over WT, left, $n = 4$) and gene expression measured by qPCR (right, $n = 4$) are shown. *$p < 0.05$; ***$p < 0.001$, one-sample (left) or two-tailed paired *t*-test (right). n.s. = non stimulated. **d** GM-CSF cultured bone-marrow derived

cells (BMDCs) were generated from WT and CD1d-KO mice. Representative flow cytometry of WT (blue) and CD1d-KO (red) BMDCs showing expression of CD1d in steady state. Secretion of IFN-β (left, $n = 3$), normalised concentration of IFN-β (fold over WT, middle, $n = 4-11$) and gene expression (right, $n = 4-8$) in WT and CD1d-KO BMDCs stimulated for 6 h as indicated (2 ng/ml LPS; 0.2 μM CpG; 5 μg/ml poly I:C). *$p < 0.05$; **$p < 0.01$; ***$p < 0.001$, 2-way ANOVA (left), one-sample (middle) or two-tailed paired *t*-test (right). n.s. = non stimulated. **e** WT mice were pre-treated with clodronate liposomes and reconstituted with WT or CD1d-KO BMDCs. Mice were injected with LPS and body temperature (left) or cytokine concentration in the blood (right) were measured ($n = 5-7$). *$p < 0.05$; **$p < 0.01$, two-tailed unpaired *t*-test. Bars in all graphs represent mean +/− SEM. Values for $n$ represent biologically independent samples (as shown by the number of data points in each graph). $n$ = cells isolated from individual mice (**a**–**d**) or mice per group (**e**). Source data are provided as a Source Data file.

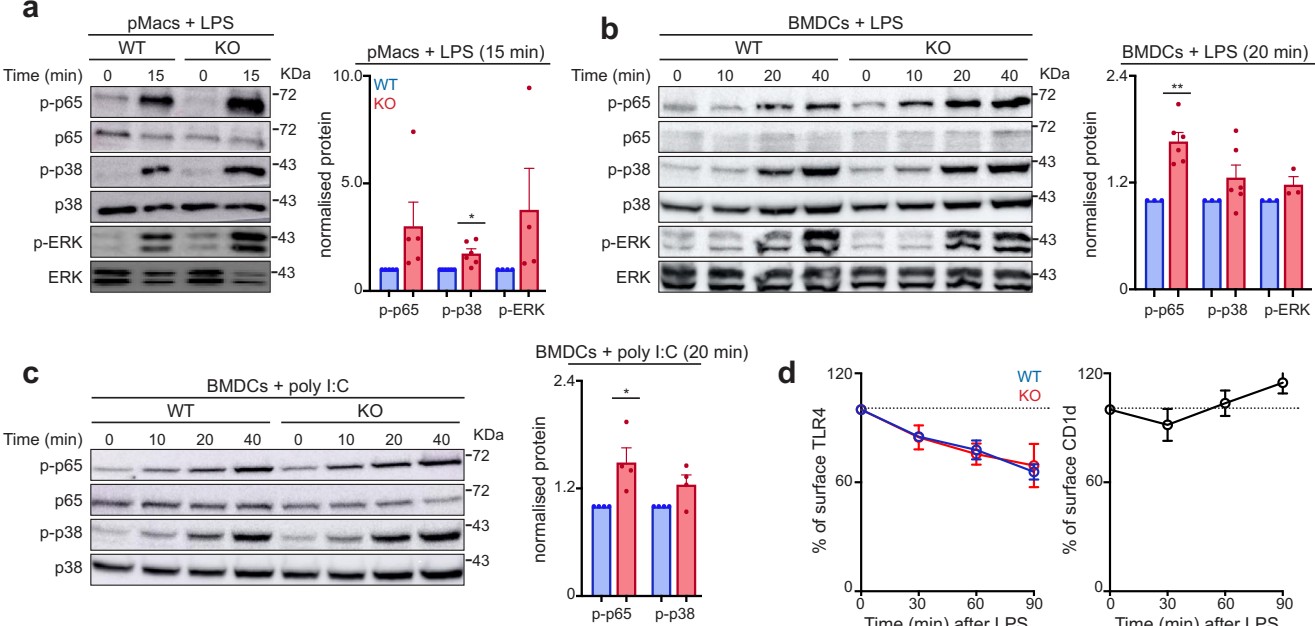

**Fig. 2 | Increased TLR signalling in CD1d-KO cells. a** Western blot analyses of phosphorylation of p65, p38 and ERK in pMacs isolated from WT or CD1d-KO mice stimulated with LPS for 15 min (left). Levels of phosphorylated p65 ($n = 5$), p38 ($n = 6$) and ERK ($n = 4$) were quantified and related to total p65, p38 and ERK respectively (right). Bars represent mean +/− SEM; *$p < 0.05$; one-sample *t*-test. **b**, **c** Western blot analyses of phosphorylation of p65, p38 and ERK in WT and CD1d-KO BMDCs stimulated with LPS (**b**) or poly I:C (**c**) for the indicated times. Levels of phosphorylated p65, p38 and ERK were quantified and related to total p65, p38 and

ERK respectively ($n = 3$–6). Bars represent mean +/− SEM; *$p < 0.05$; **$p < 0.01$, one-sample *t*-test. **d** WT and CD1dKO BMDCs were stimulated with LPS for 0, 30, 60, or 90 min and surface expression of TLR4 (left), and CD1d (right) were analysed by flow cytometry. Plots show changes in mean fluorescence intensity as a percentage of unstimulated cells (mean +/− SEM, $n = 3$). Values for $n$ represent biologically independent samples (as shown by the number of data points in each graph). $n =$ cells isolated from individual mice. Source data are provided as a Source Data file.

performed with BMDCs stimulated with either LPS or poly I:C (Fig. 2b, c). Nonetheless, CD1d deficiency did not affect surface TLR4 levels following LPS stimulation as measured by flow cytometry (Fig. 2d), suggesting that the effects in TLR signalling are independent of its internalization. Thus, these data indicate that CD1d contributes to the regulation of TLR signalling in macrophages.

**CD1d-dependent regulation of lipid metabolism in macrophages**
Next, we investigated the mechanisms underpinning the alterations in TLR responses from CD1d-KO cells. To obtain an unbiased overview of the phenotype and properties of CD1d-deficient cells, we purified pMacs from CD1d-KO mice or littermate controls and analysed their gene-expression profile by RNAseq. These analyses revealed 189 genes that were differentially expressed between WT/KO pMacs (adjusted *p*-value < 0.01, fold change >1.5), with 79 upregulated and 110 down-regulated genes in KO vs. WT cells (Fig. 3a, Supplementary Fig 3). From those, only 101 genes were mapped to protein-coding genes (Supplementary Fig 3). Gene ontology analysis using PANTHER analysis tools demonstrated a significant enrichment for genes related to lipid metabolism (Fig. 3a; note that those are the only significantly changed pathways in our dataset). Accordingly, in CD1d-KO pMacs we detected a downregulation of several genes encoding enzymes that function at different stages of the cholesterol biosynthesis pathway (including *Hmgcs1, Fdps, Cyp51, Idi1, Fdft1, Scd5*, Fig. 3a). In addition to the cholesterol pathway, the expression of genes relating to lipid metabolism were also significantly decreased in CD1d-KO vs. WT macrophages including those encoding the lipid-droplet associated protein Perilipin 1 (*Plin1*), the regulator of lipolysis Lipoprotein lipase (*Lpl*) and its co-factor *Apoc2*, as well as Phospholipid transfer protein (*Pltp*) and Sphingomyelin synthase (*Sgms1*), suggesting additional dysregulation of lipid metabolic pathways in CD1d-KO cells. We confirmed these results by qPCR, by testing the expression of enzymes of the

cholesterol pathway *Hmgcs1* (encoding Hydroxymethylglutaryl-CoA synthase) and *Cyp51* (encoding Lanosterol 14α-demethylase) as well as *Plin1, Lpl* and *Pltp* (Fig. 3b, c). These genes were found to be down-regulated in CD1d-deficient pMacs and BMDCs vs. their WT counter-parts, as well as in CD1d-KO pMacs isolated from WT:CD1d-KO mixed BM chimeras. Thus, CD1d-KO cells have cell-intrinsic alterations in lipid metabolism in steady-state conditions.

To investigate the upstream mechanisms regulating the transcriptional programme of CD1d-KO cells, we used a genome-scale mouse transcriptional regulatory network database (TRRUST)[24] to identify transcription factors upstream of genes that have significantly changed expression on CD1d-KO vs. WT cells. TRRUST predicted 2 transcription factors associated to these changes (FDR < 0.05): the Liver X receptor (LXR) family member LXRβ and the Peroxisome Proliferator-Activated Receptor (PPAR) family member PPARδ (also known as PPARβ). When we analysed the expression levels of these transcription factors in pMacs and BMDCs, we found comparable expression for *Nr1h2* (encoding LXRβ) yet the expression of *Ppard* (but not the other PPAR family members) was significantly downregulated in CD1d-KO pMacs and BMDCs (Fig. 3d–f). The downregulation of *Ppard* was also evident in CD1d-KO pMacs purified from chimeric mice (WT:CD1d-/-; 50:50) confirming that this is a cell intrinsic phenotype. Of note, other regulators of lipid metabolism including SREBP-1 (*Srebf1*), SREBP-2 (*Srebf2*) or LXRα (*Nr1h3*) showed comparable mRNA expression levels in WT and CD1d-KO cells (Fig. 3d–f).

Thus, all together these data indicate that CD1d contributes to the regulation of lipid metabolic pathways in macrophages.

**Increased lipid uptake in CD1d-KO macrophages**
Since the cholesterol biosynthesis pathway is downregulated in CD1d-KO macrophages we explored whether this alteration is translated into changes in cellular lipid content. We employed liquid

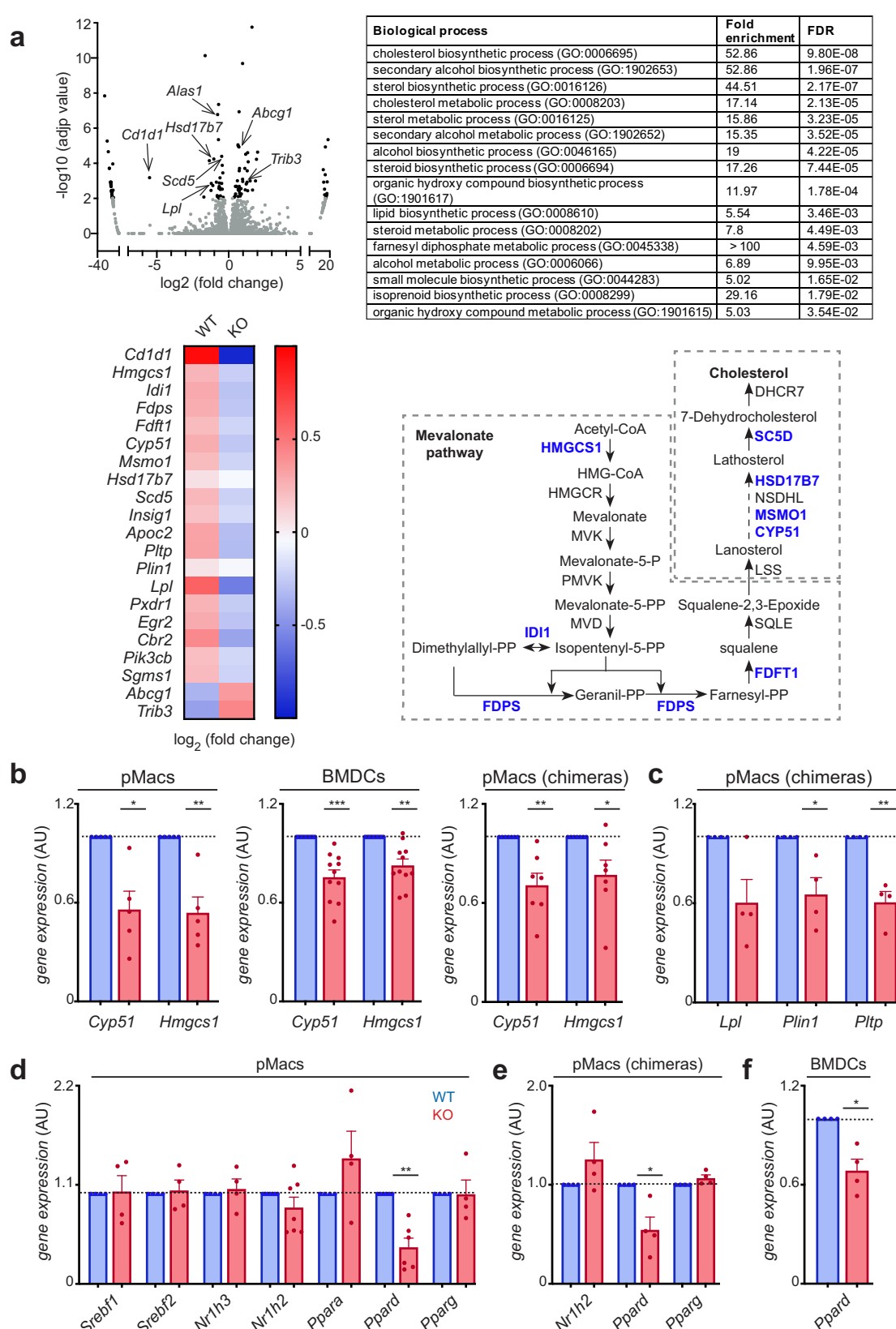

**a**

| Biological process | Fold enrichment | FDR |
|---|---|---|
| cholesterol biosynthetic process (GO:0006695) | 52.86 | 9.80E-08 |
| secondary alcohol biosynthetic process (GO:1902653) | 52.86 | 1.96E-07 |
| sterol biosynthetic process (GO:0016126) | 44.51 | 2.17E-07 |
| cholesterol metabolic process (GO:0008203) | 17.14 | 2.13E-05 |
| sterol metabolic process (GO:0016125) | 15.86 | 3.23E-05 |
| secondary alcohol metabolic process (GO:1902652) | 15.35 | 3.52E-05 |
| alcohol biosynthetic process (GO:0046165) | 19 | 4.22E-05 |
| steroid biosynthetic process (GO:0006694) | 17.26 | 7.44E-05 |
| organic hydroxy compound biosynthetic process (GO:1901617) | 11.97 | 1.78E-04 |
| lipid biosynthetic process (GO:0008610) | 5.54 | 3.46E-03 |
| steroid metabolic process (GO:0008202) | 7.8 | 4.49E-03 |
| farnesyl diphosphate metabolic process (GO:0045338) | > 100 | 4.59E-03 |
| alcohol metabolic process (GO:0006066) | 6.89 | 9.95E-03 |
| small molecule biosynthetic process (GO:0044283) | 5.02 | 1.65E-02 |
| isoprenoid biosynthetic process (GO:0008299) | 29.16 | 1.79E-02 |
| organic hydroxy compound metabolic process (GO:1901615) | 5.03 | 3.54E-02 |

chromatography-tandem mass spectrometry (LC/MS/MS) to measure cholesterol as well as cholesteryl esters (CE) in unstimulated pMacs isolated from WT and CD1d-KO mice (Fig. 4a). On a per cell basis, the levels of total cholesterol as well as those of saturated (16:0, 18:0) or unsaturated (16:1, 18:1, 18:2, 20:3, 20:4, 22:6) cholesteryl esters were not significantly different between WT and CD1d-KO pMacs. Moreover,

we did not detect any differences when staining cells with the cholesterol-binding antibiotic filipin III or when analysing cholesterol-enriched lipid rafts by staining with cholera toxin B (CtB; Fig. 4b, c); suggesting that CD1d-deficiency is not affecting cholesterol content or membrane lipid rafts in pMacs. Also, Nile Red staining -which selectively stains intracellular lipid droplets- was comparable in WT and

**Fig. 3 | Transcriptional downregulation of lipid metabolic pathways in CD1d-KO cells. a** pMacs isolated from WT or CD1d-KO mice were subjected to RNAseq analyses (*n* = 4). Top left, volcano plot including differentially expressed genes (black). A fold change cut-off of 1.5 and adjusted *p*-value cut off of 0.01 were applied. Top right, functional enrichment analysis of genes significantly changed in WT vs. CD1d-KO pMacs. The GO terms are shown ranked by *p*-values. Enrichment and *p*-values (from a Fisher's exact test with Bonferroni correction) were calculated with PANTHER tools. Bottom left, heat map for selected transcripts significantly changed in WT vs. CD1d-KO pMacs. Bottom right, representative image of the

cholesterol biosynthesis pathway, showing metabolites and enzymes responsible for the individual synthetic step. Enzymes encoded by genes significantly down-regulated in CD1d-KO pMacs are shown in blue. **b–f** qPCR analyses showing relative expression of the depicted genes in WT (blue) and CD1d-KO (red) pMacs, pMacs sort purified from 50:50 WT:CD1d-KO bone-marrow chimeras and BMDCs as indicated (*n* = 4–11). Bars represent mean +/− SEM; *\*p* < 0.05; *\*\*p* < 0.01; *\*\*\*p* < 0.001, one-sample *t*-test. Values for *n* represent biologically independent samples (as shown by the number of data points in each graph). *n* = cells isolated from individual mice. Source data are provided as a Source Data file.

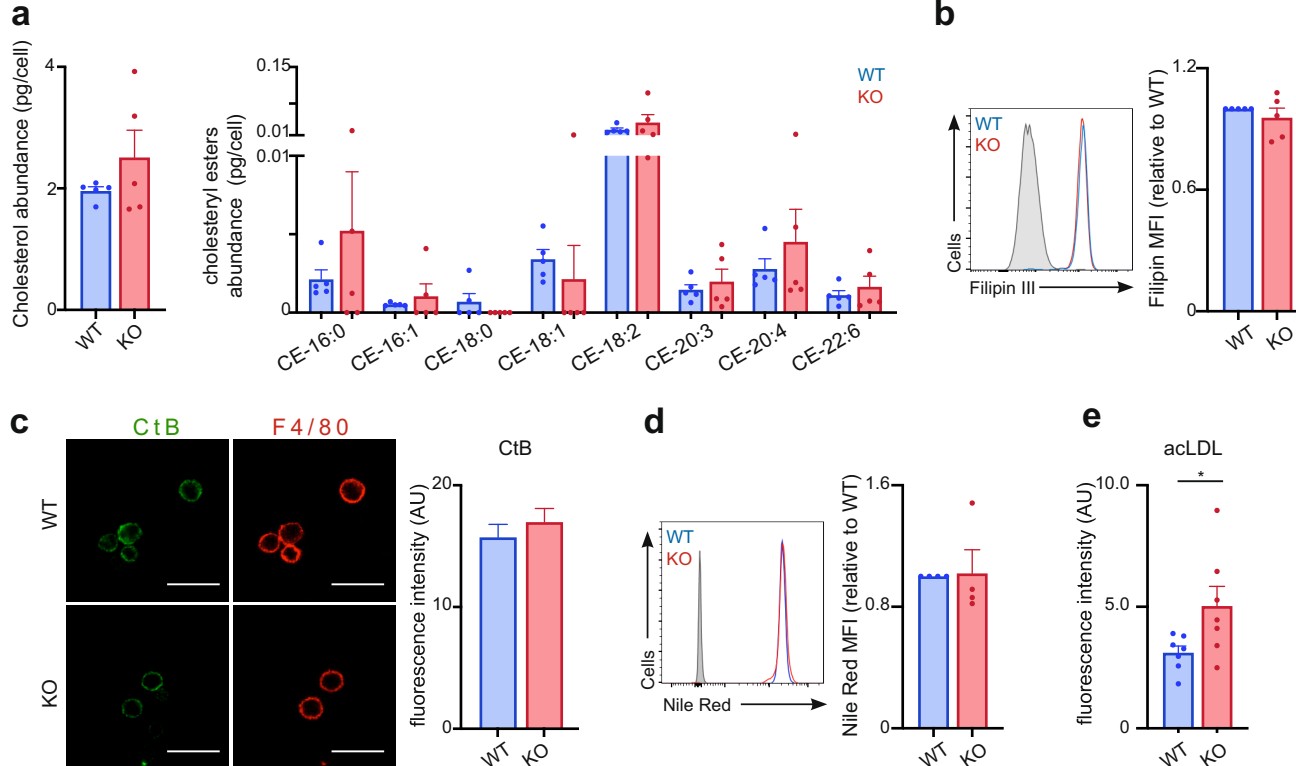

**Fig. 4 | Metabolic rewiring in CD1d-KO cells. a** Cholesterol and cholesteryl ester contents in pMacs detected using liquid chromatography-tandem mass spectrometry (LC/MS/MS) were normalized by cell numbers (*n* = 5). **b** Flow cytometry-based readout of cholesterol content by Filipin staining (*n* = 5). **c** pMacs were isolated from WT and KO mice and stained with cholera toxin B (CtB; green) and F4/80 (red). Bars represent mean +/− SEM for corrected total cell fluorescence (*n* = 161 WT; 261 CD1d-KO). Data pooled from 2 independent experiments. Scale bar is

20 μm. **d** Flow cytometry-based readout of Nile Red staining (*n* = 4). **e** Uptake of DiI-acLDL in pMacs isolated from WT or CD1d-KO mice (*n* = 7). *\*p* < 0.05, unpaired two-tailed *t*-test. Bars in all graphs represent mean +/− SEM. Values for *n* represent biologically independent samples (as shown by the number of data points in each graph). *n* = cells isolated from individual mice (**a**, **b**, **d**, **e**) or cells (**c**). Source data are provided as a Source Data file.

CD1d-KO pMacs (Fig. 4d), suggesting similar neutral lipid content in these cells.

Cellular lipid needs are strictly regulated and are achieved through a balance between de novo synthesis and import. For instance, innate signals (such as TLR stimulation) rewire macrophages' lipid metabolism by downregulating de novo fatty acid and cholesterol synthesis and upregulating lipid import, but without altering total lipid content[3,4,6]. Thus, we speculated that lipid levels in CD1d-KO cells could be balanced by an increase in lipid import. Indeed, lipid uptake -measured using fluorescently-labelled low-density lipoprotein (DiI-acetylated LDL)- was increased in CD1d-KO vs. WT pMacs in steady-state (Fig. 4e). Thus, CD1d-deficiency in macrophages rewires lipid metabolism resulting in an increase in lipid import, but without significantly altering cellular lipid availability.

**Coordinated metabolic and immune responses in macrophages**

Next, we explored the mechanism(s) underlying the metabolic rewiring in CD1d-KO macrophages by investigating the molecules

responsible for the alterations in lipid uptake. The uptake of modified lipoproteins by macrophages is largely mediated by the scavenger receptor CD36 which plays a key role in the regulation of lipid transport and cellular metabolism[25–27]. Interestingly, a recent technical report proposed the lipid transporter CD36 as a ligand for CD1 molecules demonstrating the binding of CD1 tetramers to CD36 in human monocytes, with possible implications for the functions of these proteins[28]. This prompted us to investigate whether CD36 could contribute to the increase in lipid uptake in CD1d-KO cells. To test this, we blocked CD36 in WT and CD1d-KO pMacs by incubation with sulfo-succinimidyl oleate (SSO) which irreversibly binds CD36 and has been widely used to inhibit CD36-mediated lipid import[29]. Interestingly, blocking CD36 resulted in a decrease in lipid uptake in CD1d-KO pMacs to comparable levels to WT cells (Fig. 5a), suggesting that CD36 contributes to the increase in lipid import in CD1d-KO cells. Moreover, blocking lipid uptake through CD36 was sufficient to restore the mRNA expression levels of lipid metabolic enzymes (*Hmgcs1, Cyp51, Lpl, Pltp*) which were found to be comparable in WT and CD1d-KO cells (Fig. 5b).

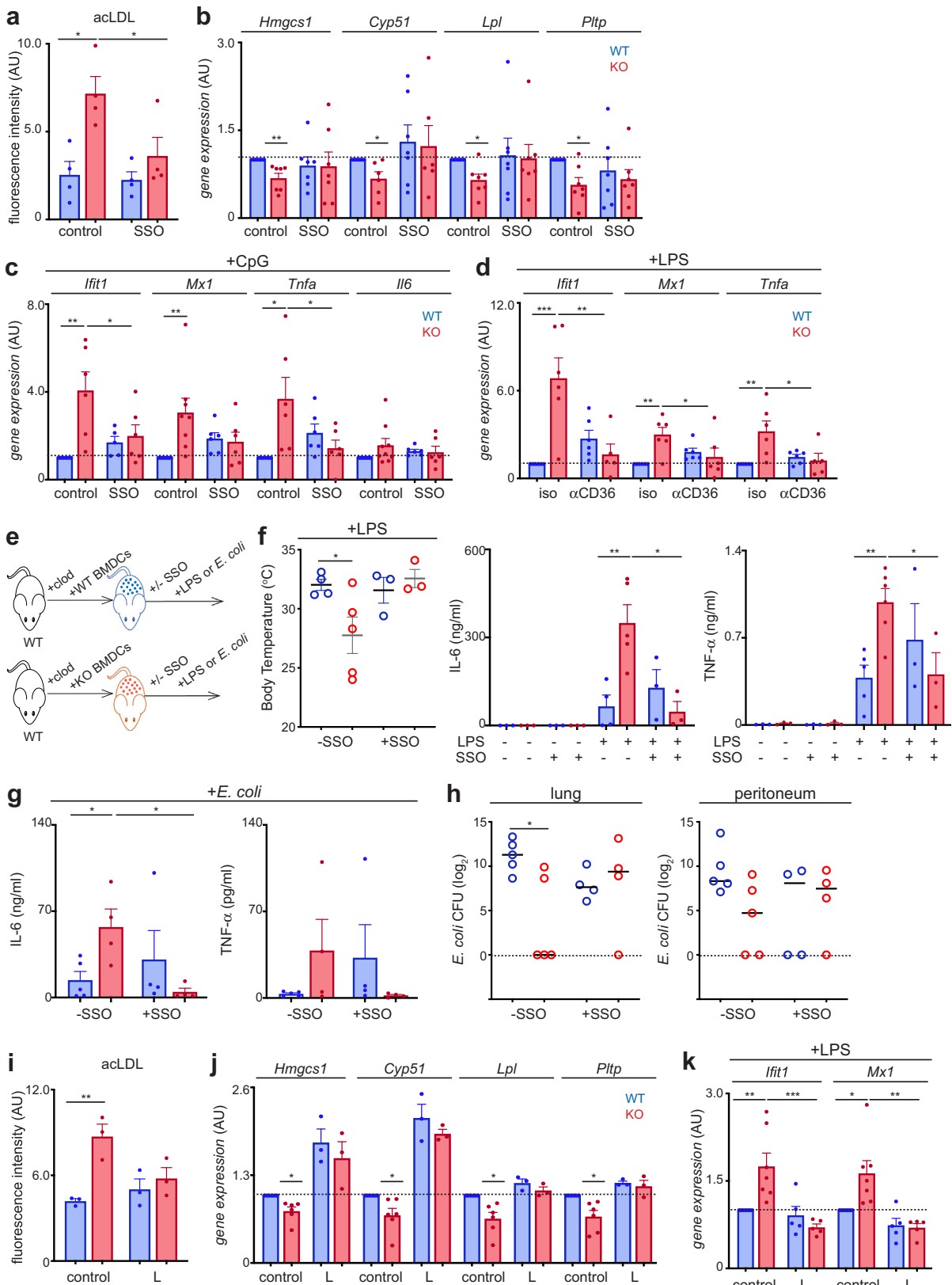

Shifts in cellular metabolism are important drivers of cell phenotype and activation[30]. Since alterations in lipid metabolic programmes are evident in unstimulated CD1d-KO cells, we hypothesised that these changes could prime cells for increased responses to TLR ligands. To functionally assess the link between the metabolic and immune alterations in CD1d-KO cells, we investigated whether restoring lipid import is sufficient to re-establish immune responses in macrophages. We used 2 approaches to block lipid import through the transporter CD36: SSO which irreversibly binds to CD36, or a CD36-blocking antibody (αCD36; JC63.1) reported to block fatty acid and modified LDL uptake[31] (Fig. 5c, d, Supplementary Fig 4a). Strikingly, blocking CD36 with SSO prior to stimulation with LPS or CpG resulted

**Fig. 5 | Lipid metabolism regulates immune responses to TLR stimulation in macrophages. a** pMacs were incubated with (or without, control) SSO and uptake of DiI-acLDL was measured ($n = 4$). *$p < 0.05$, 2-way ANOVA. **b** BMDCs were generated from WT (blue) and CD1d-deficient (red) BM and cultured in the presence (or absence, control) of SSO overnight ($n = 7$). qPCR analyses show relative expression of the depicted genes. *$p < 0.05$, **$p < 0.01$ one-sample *t*-test. **c** BMDCs were cultured in the presence (or absence, control) of SSO overnight prior to stimulation with CpG ($n = 5–9$). Expression of the depicted genes was measured by qPCR. *$p < 0.05$; **$p < 0.01$ 2-way ANOVA. **d** BMDCs were cultured in the presence of αCD36 or isotype control antibody (iso) overnight prior to stimulation with LPS ($n = 5–6$). Expression of the depicted genes was measured by qPCR. *$p < 0.05$; **$p < 0.01$; ***$p < 0.001$ 2-way ANOVA. (**e–h**) WT mice were pre-treated with clo-dronate liposomes and reconstituted with WT or CD1d-KO BMDCs ($n = 3–6$). Mice received SSO (or vehicle control) before injection of LPS (**f**) or infection with *E. coli*

(**g**, **h**). **f** Body temperature (left) or cytokine concentration in the blood (right) were measured. **g**, **h** Cytokine concentration in the blood (**g**) and bacterial CFUs (**h**) 18 h after infection. *$p < 0.05$; **$p < 0.01$, unpaired two-tailed *t*-test (**f**, **g**) or Mann-Whitney test (**h**). **i** pMacs were incubated with (or without, control) L165,041 (L) and uptake of DiI-acLDL was measured ($n = 3$). **$p < 0.01$, 2-way ANOVA. **j** BMDCs were cultured in the presence (or absence, control) of L165,041 (L) overnight ($n = 3–6$). qPCR analyses show relative expression of the depicted genes. *$p < 0.05$, one-sample *t*-test. **k** BMDCs were cultured in the presence (or absence, control) of L165,041 (L) overnight prior to stimulation with LPS ($n = 5–7$). Expression of the depicted genes was measured by qPCR. *$p < 0.05$, **$p < 0.01$, ***$p < 0.001$, 2-way ANOVA. Bars in all graphs represent mean +/− SEM, lines in **g** represent media. Values for $n$ represent biologically independent samples (as shown by the number of data points in each graph). $n$ = cells isolated from individual mice (**a-d**, **i–k**) or mice per group (**f–h**). Source data are provided as a Source Data file.

in decreased cytokine production and ISG expression in CD1d-KO cells which were detected at comparable levels to those found in WT cells (Fig. 5c, Supplementary Fig 4a). Importantly, similar results were obtained when lipid import was blocked with an αCD36 antibody (Fig. 5d), confirming that inhibition of lipid uptake through CD36 is sufficient to restore immune responses in CD1d-KO macrophages.

Next, we explored whether manipulation of metabolic pathways also regulates immune responses in vivo. To test this, we employed 2 different models of disease: an LPS-induced model of inflammation and a model of bacterial peritonitis (Fig. 5e–h). To investigate the link between inflammatory and metabolic responses, CD36 was blocked in vivo by administration of SSO to macrophage-depleted mice reconstituted with CD1d sufficient or deficient cells (Fig. 5e). As observed before (Fig. 1e), mice reconstituted with CD1d-KO cells showed increased susceptibility to LPS-induced inflammation evidenced by decreased body temperature as well as higher concentrations of cytokines in their blood (Fig. 5f). Strikingly, blocking CD36 in vivo restored immune responses, such that body temperature and blood cytokine concentration were comparable in mice reconstituted with WT and CD1d-KO cells (Fig. 5f), supporting the in vivo relevance of an immune-metabolic axis regulated by CD1d. To investigate whether this immune-metabolic axis is relevant during infection, we used a model of bacterial peritonitis caused by *Escherichia coli* as this is a clinically relevant infection with high mortality rate[32] (Fig. 5g, h). Thus, we inoculated mice intraperitoneally with $10^4$ CFUs of alive *E. coli* and checked bacterial spread and cytokine levels 18 h after infection. Mice reconstituted with CD1d-KO cells exhibited an increased concentration of cytokines in their blood and enhanced capacity to prevent bacteria dissemination as we recovered less bacterial CFUs in the lungs of those animals vs. controls. Strikingly, blocking CD36 in vivo also altered immune responses in this setting, such that cytokines and bacterial loads were comparable in mice receiving WT and CD1d-KO cells after SSO treatment (Fig. 5g, h). Thus, these data support the relevance of an immune-metabolic axis in vivo which can contribute to controlling the progression and outcome of infections.

Lastly, we investigated the molecular mechanisms linking metabolic and immune responses in macrophages. PPARδ is a ligand-inducible transcription factor with established functions in metabolism and a suggested anti-inflammatory role in immune regulation[33]. Analyses of upstream regulators of the transcriptional programme of CD1d-KO cells revealed a transcriptional down-regulation of *Ppard* (Fig. 3d–f), and western-blot analyses confirmed that total (protein) levels of PPARδ were also decreased in CD1d-KO cells (Supplementary Fig 4b). Therefore, we speculated that alterations in PPARδ may contribute to the immune-metabolic phenotype of CD1d-KO cells, and thus activation of PPARδ may be sufficient to restore macrophage responses. To test this hypothesis, we took advantage of the PPARδ agonist L165,041 which has been previously used to investigate the role of PPARδ activation in macrophage biology[34]. Interestingly, PPARδ activation was sufficient to decrease acLDL uptake in CD1d-deficient

cells (Fig. 5i) and restored the expression of lipid metabolic genes to comparable levels to WT cells (Fig. 5j). Moreover, activation of PPARδ prior to stimulation with LPS resulted in decreased ISG expression in CD1d-KO cells comparable to WT cells (Fig. 5k). Thus, activation of PPARδ restores metabolic and immune responses in CD1d-KO cells.

Altogether these data indicate that the CD1d-dependent metabolic reprogramming in macrophages contributes to the regulation of immune responses.

## CD1d modulates CD36 internalization and lipid uptake

Finally, we explored the molecular mechanisms by which CD1d modulates lipid uptake. First, we investigated whether altered expression and/or function of CD36 could be responsible for the changes in lipid import in CD1d-KO cells. qPCR analyses showed comparable *Cd36* (and scavenger receptor *Msr1*) mRNA levels in WT and KO pMacs and BMDCs (Fig. 6a), suggesting that CD1d-deficiency does not affect their expression. Moreover, total levels of CD36 were found to be comparable in WT and CD1d-KO cells as detected by western blot (Fig. 6b). However, flow cytometry staining showed significantly decreased surface levels of CD36 in CD1d-KO cells, yet the levels of MSR1 are comparable (Fig. 6c). We confirmed this result by confocal microscopy, which showed both CD1d and CD36 at the cell membrane in WT cells, yet we detected a notable decrease in surface CD36 fluorescence intensity in CD1d-KO cells (Fig. 6d), suggesting that CD1d may modulate the location of CD36 at the plasma membrane. Interestingly, analyses of confocal microscopy images showed a degree of colocalization between CD1d and CD36 on the cell surface (Fig. 6d), suggesting that these molecules are located in close proximity at the plasma membrane. In line with these data, proximity ligation assays (PLA) indicate that CD1d and CD36 are found in close proximity on the cell surface as we detected CD1d-CD36 PLA puncta in WT cells which were absent in CD1d-KO cells (Fig. 6e). Thus, these data indicate that CD1d does not affect CD36 expression, yet it regulates its localization at the plasma membrane.

The internalization of CD36 is a crucial step for lipid uptake and this process is modulated by a variety of membrane receptors and adaptor proteins[35–37]. For example, the adhesion molecule CD146 (MCAM) regulates CD36 internalization in macrophages and subsequent lipid uptake[35]. Thus, we speculated that CD1d may regulate CD36 internalization consequently controlling lipid internalization. To test this hypothesis, we incubated WT and CD1d-KO pMacs with oxidated-LDL (oxLDL) particles which have been shown to induce internalization of the CD36-oxLDL complex (Fig. 6f). Flow cytometry analysis of CD36 showed a decrease in the levels of surface CD36 after oxLDL treatment, yet this decrease was significantly higher in CD1d-KO cells, indicating an increased level of CD36 internalization in the absence of CD1d (Fig. 6f). To further validate the role of CD1d in controlling CD36-mediated lipid uptake, we incubated WT pMacs with a CD1d blocking antibody (19G11[15]) and measured the effect in lipid uptake as well as in the subsequent immune responses after LPS

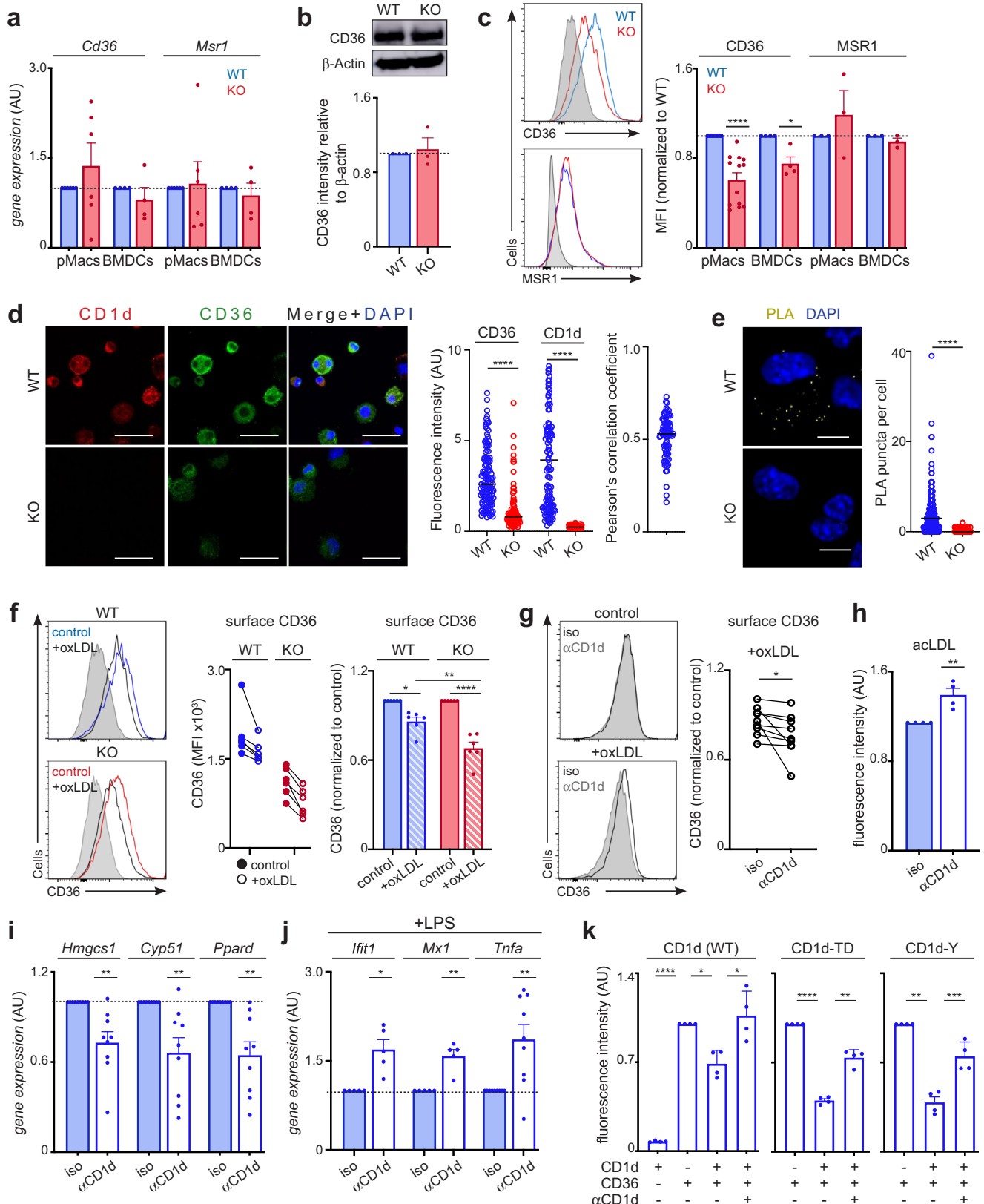

stimulation. Strikingly, blocking surface CD1d in WT pMacs was sufficient to induce a small but significant increase in CD36 internalization after oxLDL treatment (Fig. 6g), which was accompanied by an increase in lipid uptake as measured using fluorescently-labelled acLDL (Fig. 6h). Moreover, blocking CD1d in WT cells resulted in a down-regulation of lipid metabolic genes (*Hmgcs1*, *Cyp51*) as well as *Ppard*

(Fig. 6i), and a concomitant increase in cytokine/ISG production in response to TLR stimulation (Fig. 6j). Thus, blocking surface CD1d is sufficient to recapitulate the metabolic and immune responses found in CD1d-deficient cells. Comparable results were obtained when blocking CD1d in RAW264.7 macrophages (which express high levels of endogenous CD1d and CD36). CD1d-blocking resulted in altered

**Fig. 6 | CD1d regulates CD36 internalization and lipid uptake. a** qPCR expression data of *Cd36* and *Msr1* in WT and CD1d-KO cells (*n* = 4–6). **b** Western-blots and quantification for CD36 and β-actin in WT and CD1d-KO BMDCs (*n* = 3). **c** Flow cytometry images and quantification of Mean fluorescence intensity (MFI) for CD36 and MSR1 in WT and CD1d-KO cells (*n* = 3–13). *$p < 0.05$, ***$p < 0.001$, one-sample *t*-test. **d** Confocal microscopy images showing staining of CD1d (red), CD36 (green) and DAPI (blue) for WT (*n* = 123) and CD1d-KO BMDCs (*n* = 91). Graphs show fluorescence intensity for CD1d and CD36 (left) and Pearson correlation coefficient (right). Scale bar is 20 μm. ****$p < 0.0001$ two-tailed unpaired *t*-test. **e** PLA was performed in WT and CD1d-KO BMDCs for CD1d and CD36. Representative images of PLA (yellow) and DAPI (blue) and quantification of PLA (puncta per cell) are shown (*n* = 162–305 cells). Scale bar is 7 μm. ****$p < 0.0001$ two-tailed unpaired *t*-test. **f** Flow-cytometry analyses of surface CD36 in WT and CD1d-KO pMacs after incubation with oxLDL (left). MFI for CD36 (middle) or values relative to control (untreated) cells (right) are shown (*n* = 6). **$p < 0.01$; ***$p < 0.001$; ****$p < 0.0001$, 2-way ANOVA. **g** Flow-cytometry analyses of CD36 in WT pMacs pre-treated with αCD1d or isotype (iso) and incubated with oxLDL (*n* = 9). CD36 levels are shown relative to control (untreated cells + isotype). *$p < 0.05$; paired two-tailed *t*-test. **h** Uptake of DiI-acLDL in pMacs pre-treated with αCD1d or isotype (*n* = 4). **$p < 0.01$, unpaired two-tailed *t*-test. **i** BMDCs were cultured with αCD1d or isotype. qPCR data show relative expression of the depicted genes (*n* = 9). **$p < 0.01$ one-sample *t*-test. **j** BMDCs were cultured with αCD1d or isotype (iso) and stimulated with LPS (*n* = 5–9). Expression of the depicted genes was measured by qPCR. *$p < 0.05$, **$p < 0.01$ one-sample *t*-test. **k** Uptake of DiI-acLDL in HEK293T cells transfected with CD1d, mutant CD1d (CD1d-TD or CD1d-Y) and/or CD36 (*n* = 4). Some samples were incubated with αCD1d as indicated. *$p < 0.05$; **$p < 0.01$; ***$p < 0.001$, one-way ANOVA. Bars in all graphs represent mean +/− SEM. Values for *n* represent biologically independent samples (as shown by the number of data points in each graph). *n* = cells isolated from individual mice (**a–c**, **f–j**), cells (**d**, **e**) or transfections (**k**). Source data are provided as a Source Data file.

metabolic and immune responses in these cells (Supplementary Fig 5a–d). Hence, these data further support that CD1d modulates lipid internalization through CD36 leading to the subsequent regulation of metabolism and immunity.

Finally, we explored the molecular requirements by which CD1d controls CD36-dependent lipid uptake. To do this, we set up a system in which we expressed CD1d and/or CD36 in HEK293T cells and measured uptake of fluorescent acLDL (Fig. 6k, Supplementary Fig 5e). We detected a stark increase in lipid uptake following expression of CD36, which was not detected in cells expressing only CD1d (or in mock transfected cells), indicating that internalization of acLDL in this system is primarily mediated by CD36. Co-expression of CD1d and CD36 resulted in a decrease in lipid internalization compared to CD36-expressing cells, and this effect was reverted by incubation of cells with an αCD1d blocking antibody, confirming that also in this system CD1d modulates CD36-mediated lipid import. CD1d has a short intracellular tail which has been suggested to participate in retrograde signalling and to control CD1d internalization and trafficking[11,38,39]. To investigate whether the CD1d tail plays a role in the regulation of lipid uptake, we expressed two mutant forms of CD1d either lacking 7 residues (SAYQDIR) of the intracellular tail (CD1d-TD) or with a Y → A mutation in Y332 (CD1d-Y). Both of these mutants were efficiently expressed and detected at the cell surface (Supplementary Fig 5e) as previously reported[38,39]. Interestingly, expression of mutant CD1d (CD1d-TD or CD1d-Y) in CD36-expressing cells induced a decrease in lipid uptake which was reverted by incubation with an αCD1d antibody (Fig. 6k), suggesting that the CD1d-dependent regulation of lipid uptake by CD36 is independent of the CD1d cytoplasmatic tail.

Thus, collectively our data demonstrate that CD1d modulates CD36-mediated lipid uptake and subsequent metabolic and immune responses in macrophages.

## Discussion

Alterations in cellular metabolism are important drivers in the control of macrophage phenotype and activation. In this paper, we identify the lipid presenting molecule CD1d as a regulator of lipid metabolism in macrophages ultimately controlling macrophage response to innate stimuli (Supplementary Fig 6). We show that absence of CD1d induces a rewiring in cellular lipid metabolism underpinned by an increase in lipid import. These metabolic alterations shape macrophage response to innate stimuli, as CD1d-deficient cells show increased activation in response to TLR stimulation. Mechanistically, CD1d regulates the internalization of CD36 ultimately controlling lipid uptake; while blocking lipid import through CD36 restores metabolic and immune responses in macrophages. Thus, our data identifies a previously unrecognised function for CD1d as a link between lipid metabolism and innate immunity in macrophages.

The better understood function of CD1d relates to its capacity to present lipid antigens to NKT cells. However, an increasing body of

data has shown cell intrinsic functions for CD1d controlling cytokine secretion in hematopoietic and non-hematopoietic cells. For instance, in intestinal epithelial cells, ligation of CD1d results in STAT-3 dependent secretion of IL-10[15]; while CD1d crosslinking in human monocytes leads to IL-12 secretion in a process mediated by NF-kB[12]. On the other hand, recent publications reported controversial results regarding the role of CD1d in the (positive or negative) regulation of inflammation[14,18]. Using two independently generated CD1d-KO strains bred in two separate animal facilities, our data demonstrates that CD1d negatively regulates TLR signalling, thus controlling the production of proinflammatory cytokines and type-I IFN. This function of CD1d is cell intrinsic and independent of NKT cells. Interestingly, the levels of expression of CD1d have been shown to be altered in a variety of pathologies including autoimmune diseases such as systemic lupus erythematosus (SLE)[16]. Whilst it is likely that alteration of CD1d expression will in turn control NKT cell functions in the context of disease, it is also possible that the downregulation of CD1d itself could contribute to the control of TLR signals and consequently shape the development of the inflammatory response.

Our transcriptome analyses revealed a downregulation of lipid metabolic pathways in CD1d-deficient macrophages, which occurred in a cell-intrinsic manner and in the absence of stimulation. This effect was accompanied by an increase in lipid import, suggesting that CD1d deletion induces a metabolic rewiring in macrophages favouring reliance on exogenous lipid uptake over de novo synthesis. Shifts in cellular metabolism are known to be important drivers of cell phenotype and activation[30]. Indeed, our results show that manipulation of lipid pathways is sufficient to restore the hyperactivated TLR responses in CD1d-KO cells. These data are in line with recent reports showing that TLR responses and lipid metabolism are connected enabling immune cells to coordinate metabolic changes with immune activation. For instance, genetic disruption of enzymes of the cholesterol pathway is sufficient to shift the balance between cholesterol synthesis and import, resulting in increased cytokine production in response to TLR stimulation[4]. Conversely, immune signals (such as TLR ligands or type I IFN) lead to downregulation of fatty acid and cholesterol synthesis without affecting cellular lipid content[3,4,40]. Thus, these reports define a metabolic-inflammatory circuit in myeloid cells by which lipid biosynthesis pathways can be monitored by the host defence machinery controlling responses to external insults. In line with this model, we propose that the metabolic rewiring induced by the lack of CD1d, primes cells for hyper-responsiveness to TLR ligands. While the mechanism(s) linking lipid metabolism and innate signals remain poorly defined, it has been proposed that differential subcellular concentrations of synthesised vs. imported lipids or their metabolites (e.g. endoplasmic reticulum (ER) vs. plasma membrane) could convey distinct information to the cells regarding lipid homoeostasis and regulate the activation of signalling molecules in the ER[4]. On the other hand, downregulation of lipid biosynthesis pathways can lead to the

reduction or accumulation of key metabolites of the pathways which can in turn control macrophage functions[8,41].

Our data shows that CD1d contributes to the regulation of lipid metabolism by controlling CD36 function. The CD36-dependent increase in lipid import in CD1d-KO cells was not due to alterations in CD36 expression, but related to the altered internalisation of CD36, suggesting that CD1d controls lipid import through regulation of CD36 internalization. Moreover, both molecules are located in membrane lipid rafts[42,43], suggesting that they cooperate in specific membrane microdomains to ultimately regulate lipid uptake. In line with this, other surface molecules -such as CD146- have been shown to modulate CD36 internalization during lipid uptake ultimately controlling macrophage activation[35]. Interestingly, a recent technical report has identified CD36 as a ligand for human CD1 tetramers[28]. While the physiological relevance of this observation remains unexplored, the authors suggest that CD36 could play a role in the pathways of lipid loading into CD1 molecules. Our data further support a CD1d-CD36 axis by which CD1d could in turn modulate the internalization of CD36 and consequently control lipid uptake. It is worth noting that a similar function could be also relevant to other CD1 family members expressed in human cells, such as CD1b or CD1c, as tetramers from these molecules have also been shown to bind CD36 in human monocytes[28]. CD1d molecules have a short intracellular tail which has been shown to control their intracellular trafficking and rate of internalization in steady state[38,39]. However, our data indicate that the regulation of CD36-mediated lipid uptake is independent of the CD1d intracellular tail, as CD1d-tail mutants also modulate CD36 function. Moreover, lipid internalization can be altered by incubation of cells with an αCD1d antibody, suggesting that surface CD1d -rather than the intracellular portion- may be important for this function. The mechanism(s) by which CD1d modulates CD36-mediated lipid uptake will require further investigation, but it may comprise a variety of factors including regulation of intracellular trafficking, retention at the plasma membrane and/or competition for lipid binding.

Our data indicate that CD36 contributes to the metabolic and immune alterations of CD1d-KO macrophages as inhibition of lipid import through CD36 restores lipid uptake and TLR responses in CD1d-KO cells. CD36 integrates cell signalling and metabolic pathways such that CD36-dependent metabolic rewiring has been shown to regulate the activation and function of a variety of cells[26,27,44]. In macrophages, CD36 has been proposed to promote activation and cytokine secretion and it can also control macrophage polarization[44]. CD36 can bind a variety of lipids as well as damage-associated molecular patterns (DAMPs) including oxLDL, amyloid β proteins or glycated proteins. CD36-dependent internalization of long-chain fatty acids or oxLDL by macrophages induces a metabolic switch, that ultimately leads to pro-inflammatory cytokine secretion[44,45]. In line with these reports, we found that increased CD36-mediated lipid uptake in CD1d-KO macrophages results in increased inflammatory responses, which are restored after CD36 blocking. It is important to note that CD1d and CD36 are both expressed by a variety of hematopoietic and non-hematopoietic cells (IECs, adipocytes, hepatocytes) so it is likely that the macrophage CD1d-CD36 cross-regulation reported here may have broader physiological implications in different cell types, tissues and disease contexts. One of such cases, could be in tumour progression. Surface CD1d expression is lost or downregulated in the majority of solid tumours and in some cancers -such as multiple myeloma- CD1d downregulation correlates with increased metastatic potential and disease progression[46]. Whilst it is likely that alteration of CD1d expression will in turn enable tumour escape from NKT cell-mediated immunosurveillance, it is also possible that altered expression of CD1d contributes to metabolic alterations in tumour cells by regulating CD36 functions. In line with this, metastasis-initiating cells have been described to express high levels of CD36, and their metastatic potential can be blocked with αCD36 neutralising antibodies[31]. Moreover,

CD36-mediated metabolic rewiring of breast cancer cells has been shown to promote resistance to therapy[26]. Thus, it is tempting to speculate that CD1d-dependent regulation of CD36 functions in cancer cells could contribute to modulate their survival, proliferation and/or metastatic potential and consequently influence disease progression and response to treatment.

Analyses of upstream regulators of the transcriptional programme of CD1d-KO cells revealed a specific down-regulation of the transcription factor PPARδ. PPARδ is a lipid-activated nuclear receptor which has critical functions in the regulation of lipid homoeostasis and has been proposed to play an anti-inflammatory role in immunity[33]. In macrophages, PPARδ regulates M2 polarization[47,48], and myeloid-deletion of *Ppard* causes increased inflammation in adipose tissue in response to high-fat diet[47]. Conversely, PPARδ agonists dampen inflammatory responses in models of dermal wound healing or intestinal inflammation[49,50]. In human monocytes, activation of PPARδ with the agonist L165,041 shows immune-suppressive effects, inducing the repression of inflammation-associated NFκB and STAT1 target genes[34]. In keeping with these data, activation of PPARδ in CD1d-KO cells restores immune responses, suggesting that PPARδ function contributes to the CD1d-dependent regulation of immunity. PPARδ is ubiquitously expressed and its function is known to be regulated in a ligand-dependent manner. Different molecules have been proposed as natural ligands for PPARδ including unsaturated fatty acids, saturated fatty acids, eicosanoids and very low-density lipoprotein-derived fatty acids[51,52]. Interestingly, CD1d-dependent alterations in lipid uptake are sufficient to induce a down-regulation of *Ppard* expression, suggesting that differential uptake/accumulation of lipids in CD1d-deficient cells (or in response to CD1d-blockage) may be sufficient to alter *Ppard* expression (and possibly its function), consequently controlling immune responses.

The role of CD1d in the regulation of lipid metabolism described here, can have broad implications for metabolic diseases including obesity or atherosclerosis. While the role of CD1d/NKT cells in obesity and metabolic disorders remains unclear, some studies have found that CD1d-KO mice fed a high-fat diet have increased steatosis and impaired hepatic glucose tolerance[53,54]. Moreover, the macrophage population is altered in the adipose tissue of CD1d-KO mice with an increase on M1-CD11c+ and iNOS+ macrophages in CD1d-deficient animals in steady state conditions[55]. Since CD1d-KO mice lack NKT cells, it is not possible to differentiate the contribution of CD1d itself vs. NKT cells in these phenotypes. However, some of the metabolic alterations found in CD1d-KO mice were not recapitulated (or found to be milder) in Jα18-KO mice (lacking invariant NKT cells)[54], suggesting a role for CD1d in metabolic regulation which may happen independently of iNKT cells.

In conclusion, we define a previously unrecognised function for CD1d in the regulation of lipid metabolism in macrophages which in turn controls responses to innate signals. This function is cell-intrinsic and independent of CD1d's role in the regulation of T cell immunity. Given the broad cellular expression of CD1d, the metabolic function described here will have wide implications for immune regulation in homoeostasis as well as in metabolic, infectious and autoimmune diseases.

## Methods

### Mice and cell lines

CD1d-KO (B6.129S6-Del(3Cd1d2-Cd1d1)1Sbp/J and CD1d^flox x PGK^Cre), WT C57BL/6, congenic CD45.1 or CD45.1/CD45.2 WT C57BL/6 mice were bred and maintained in individually ventilated cages under specific pathogen-free conditions at the Francis Crick Institute or King's College London. All mice were housed under a 12-h light/12-h dark cycle with ad libitum access to food and water, at a temperature of 19–21 °C and humidity of 45–65%. Age- and sex-matched mice between 8 and 16 weeks of age were used in the experiments. All animal

experiments were approved by the Francis Crick Institute and the King's College London's Animal Welfare and Ethical Review Body and the United Kingdom Home Office.

HEK293T and RAW264.7 cells were obtained from The Francis Crick Institute Cell Services and maintained in complete DMEM media (Gibco; 10% FCS, 100U/ml penicillin, 100 μg/ml streptomycin, 14.3 μM β-Mercaptoethanol).

## Generation of BMDCs and isolation of pMacs

Bone marrow cells were flushed from the femurs and tibias with cold PBS, filtered through a 45 μm strainer, pelleted by centrifugation and resuspended in complete RPMI media (Gibco; 10% FCS, 100 U/ml penicillin, 100 μg/ml streptomycin, 14.3 μM β-Mercaptoethanol) supplemented with 20 ng/ml of GM-CSF (Biolegend). Medium was changed at days 3 and 4 of culture and cells were harvested on day 6. For stimulation experiments, $CD11c^+$ cells were enriched by positive selection with CD11c magnetic beads (CD11c MicroBeads UltraPure, Miltenyi).

To isolate pMacs, mice were sacrificed using $CO_2$ and 4 ml of ice-cold PBS injected into the peritoneal cavity. The peritoneal cavity was briefly massaged and cells recovered using a 10 ml syringe. For stimulation experiments pMacs were enriched by positive selection with F4/80 magnetic beads (Anti-F4/80 MicroBeads UltraPure, Miltenyi).

## Flow cytometry

Flow cytometry staining were performed in FACS buffer (PBS 1% BSA, 1% FCS) using the following anti-mouse antibodies from Biolegend unless specified otherwise (at a dilution of 1:200): CD1d (1B1), CD11b (M1/70), CD11c (N418), CCR7 (4B12), CD45.1 (A20), CD45.2 (104), CD24 (M1/69), CD40 (3/23), CD117 (2B8), CD86 (GL-1), TLR4 (SA15-21), MHC-I (AF6-88.5, BD Biosciences), MHC-II (M5/114.15.2), CD36 (HM36), MSR1 (1F8C33), CD80 (16-10A1), CD69 (H1.2 F). Dead cells were excluded from the analyses by staining with DAPI or Zombie fixable viability dye (Biolegend). Flow cytometry data were collected on an LSR-II or LSRFortessa (both from BD Biosciences) using FACSDiva Software and were analysed with FlowJo software (TreeStar).

Cholesterol staining with Filipin III was performed using Cholesterol Assay Kit (Abcam) and lipid staining was performed using Nile Red (ThermoFisher) according to the manufacturer's instructions. For CD36 internalization, pMACs were treated with 2 μg/ml αCD1d (19G11, InVivoMab) or isotype control for 1 h, prior to stimulation with 250 μg/ml oxLDL (ThermoFisher) for a further 1 h.

## Generation of bone-marrow chimeras and in vivo models of disease

To generate bone-marrow chimeras, lethally irradiated recipient mice ($CD45.1^+CD45.2^+$) were adoptively transferred with bone marrow from WT ($CD45.1^+$) and CD1d-KO ($CD45.2^+$) donor mice mixed in a 50:50 ratio. Six weeks post transfer, pMacs were purified by cell sorting (with a FACSAria II; BD Biosciences) on the basis of congenic marker expression as $CD11b^+F4/80^+CD45.1^+$ or $CD11b^+F4/80^+CD45.2^+$ cells and used for further experiments.

For in vivo models of disease, WT mice were intraperitoneally (i.p.) injected with 200 μl of clodronate liposomes (Liposoma) and 3 days later were adoptively transferred with $3 \times 10^6$ WT or CD1d-KO BMDCs. Mice were injected i.p. with 40 mg/kg of SSO or vehicle control 4 h prior to LPS or E. coli injection. For induction of LPS-induced inflammation, mice were injected i.p. with 5 μg/g of LPS or vehicle control. Mice were bled via tail prick immediately prior to LPS injection and 4 h post injection. Temperature was measured using a rectal probe at the same time-points. For induction of peritonitis, mice were injected i.p. with $10^4$ colony forming units (CFUs) of live E. coli (DH5α). Mice were bled via tail prick 18 h post E. coli injection and organs were collected and plated for CFU counts.

## Stimulation and pharmacological modulation of BMDCs, pMacs and RAW264.7 cells

50,000–100,000 pMACs or 100,000 BMDCs or RAW264.7 cells were plated in 100 μl complete media in 96 well plates. Cells were stimulated with the TLR ligands LPS (derived from Escherichia coli K12; LPS-EK, Invivogen), PolyI:C (Sigma), CpG (class C ODN 2395, Miltenyi) or alive E. coli (DH5α) at the indicated concentrations. Where stated, cells were cultured overnight with 25 μM SSO (Abcam), 5 μg/ml CD36 blocking antibody (JC63.1, Abcam), 2 μg/ml αCD1d (19G11, InVivoMab) or isotype control, or 10 μM L165,041 (Tocris) prior to stimulation. BMDCs were also stimulated for 6 h with 0.1 ng/ml IFN-β (Biolegend), 0.05 ng/ml TNF-α (Biolegend) or 20 ng/ml IFN-γ (Biolegend). Cytokines were measured in the supernatant of the culture by cytometric bead array (LEGENDplex™ Mouse Inflammation Panel, Biolegend) and mRNA was measured by qPCR.

## Western-blot

To analyse protein phosphorylation, 3–5 million BMDCs or 2 million pMacs were stimulated with LPS or poly I:C as indicated and cells lysed in lysis buffer (150 mM NaCl, 50 mM TrisHCl pH8.0, 0.1% SDS, 0.5% Sodium Deoxycholate, 1% Triton) supplemented with protease (cOmplete, Sigma) and phosphatase (PhosSTOP, Sigma) inhibitors. Samples were resolved by SDS-PAGE and proteins transferred to PVDF membranes. The following antibodies were used for western blot (all from Cell Signalling unless specified otherwise): β-actin (1:3000, 8H10D10), ERK (1:1000, L34F12), p-ERK (1:1000, 9101), p38 (1:1000, D13E1), p-p38 (1:1000, 28B10), p65 (1:1000, D14E12), p-p65 (1:1000, 93H1), CD36 (1:1000, PA5-33291, ThermoFisher), anti-mouse IgG (1:3000, Poly4053, Biolegend), anti-rabbit IgG (1:3000, sc-2955, Santa Cruz Biotechnology), PPARδ (1:500, ab23673, Abcam). Membranes were imaged in an Amersham Imager 600 and band intensity quantified with Fiji (ImageJ).

## qPCR

RNA extraction was performed using RNeasy Mini Kit (Qiagen). cDNA was synthesised with iScript Select cDNA Synthesis Kit (Bio-Rad) and gene expression was determined with iTaq Universal SyBR Green Supermix (Bio-Rad) and the primers included in Supplementary Table 1. Reactions were run in a real-time PCR system (ABI7900HT; Applied Biosystems).

## Lipidomics

pMacs were isolated from WT and CD1d-KO mice with F4/80 magnetic beads, counted, pelleted and flash-frozen before lipid extraction. Samples were spiked with 10 μl splash mix (Avanti) prior to lipid extraction. Lipids were extracted by adding a 2.5 ml solvent mixture (1 M acetic acid/isopropanol/hexane; 2:20:30, v/v/v) to 1 ml cells followed by 2.5 ml hexane. Tubes were centrifuged (500 g for 5 min at 4 °C) to recover lipids in the upper hexane layer (aqueous phase). Aqueous samples were re-extracted as above by addition of 2.5 ml hexane, and upper layers were combined. Lipid extraction from the lower aqueous layer was then completed according to the Bligh and Dyer technique. Specifically, 3.75 ml of a 2:1 ratio of methanol:chloroform was added followed by vortexing. Subsequent additions of 1.25 ml chloroform and 1.25 ml water were followed with a vortexing step, and the lower layer was recovered following centrifugation as above and combined with the upper layers from the first stage of extraction. Solvent was dried under vacuum and lipid extract was reconstituted in 100 μl HPLC grade methanol. LC-MS/MS for free cholesterol and cholesterol esters was performed on a Nexera liquid chromatography system (Shimadzu) coupled to an API 4000 qTrap mass spectrometer (Sciex). Liquid chromatography was performed at 40 °C using a Hypersil Gold C18 (Thermo Fisher Scientific) reversed phase column (100 × 2.1 mm, 1.9 μm) at a flow rate of 0.4 mL/min over 11 min. Mobile phase A was (water/solvent B 95/5; v/v and 4 mM ammonium acetate) and mobile phase B was acetonitrile/isopropanol

(60/40; v/v and 4 mM ammonium acetate). The following linear gradient for B was applied: 90% for 1 min, 90–100% from 1 to 5 min and held at 100% for 3 min followed by 3 min at initial condition for column re-equilibration. Free cholesterol and CEs were analysed in MRM mode monitoring the parent to daughter transitions of 12 CEs and free cholesterol. Cholesterol and CEs were quantified using an external calibration with the following standards: Cholesterol and Cholersterol-d7, CE(14:0), CE(16:0), CE(18:0), CE(18:1), CE(20:4), CE(22:6) and CE(18:1-d7). Inclusion criteria for chromatographic peaks were those with a signal to noise ratio of at least 5:1 and at least 7 points across the peak.

## Lipid uptake
Dil-acLDL uptake was measured as described[4]. In brief, cells were plated in lipid-free medium in the presence or absence of 25 µM SSO (for 4 h), 2 µg/ml αCD1d or isotype control (for 1 h), or 10 µM L165,041 (overnight) prior to the addition of 5 µg/ml of Dil-acLDL (Invitrogen) for a further 2 h. Cells were washed and counted and $1.5×10^5$ cells per sample were lysed in RIPA Buffer. 20 µL of lysate was transferred to dark-walled, clear bottom 384 well dish and fluorescence intensity was measured with an EnSight Mulitmode Microplate Reader (PerkinElmer).

## Constructs and transfections
The plasmid containing mouse CD36 (pCMV3-HA) was purchased from SinoBiological. Mouse CD1d (containing a FLAG-tag in the C-terminus) was cloned into a pCMS28 vector (kindly donated by Dr Monica Agromayor, King's College London). Tail deleted and $Tyr^{332} \rightarrow Ala$ (Y332A) CD1d mutants (CD1d-TD and CD1d-Y, respectively) were generated by site directed mutagenesis using the Q5 Site-Directed Mutagenesis Kit (New England Biolabs) according to the manufacturer's instructions. HEK293T cells were transfected with CD36 and/or CD1d (WT, TD or Y) plasmids, using polyethylenimine following standard techniques and cells were harvested for analyses 48 h after transfection.

## Microscopy
For confocal microscopy stainings for CD1d and CD36, BMDCs were immobilised onto coverslips by cytospin centrifugation, fixed with 4% paraformaldehyde, and stained with anti CD1d (1:100, 1B1, Biolegend) and anti-CD36 (1:100, EPR6573, Abcam) followed by anti-rat-Alexa555 and anti-rabbit-Alexa 647 secondary antibodies (1:500). Nuclei were stained with DAPI.

For lipid raft staining, pMacs were plated on glass cover slips and left to adhere for 2 h. Cells were washed once with PBS, and lipid rafts stained according to manufacturer's instructions using the Vybrant Lipid Raft Labeling Kit (Invitrogen). Briefly, cells were incubated for 10 min with Alexa488-conjugated cholera toxin subunit B (CTB) at 4 °C, followed by cross-linking with an anti-CTB antibody for 15 min at 4 °C. Cells were washed twice in cold PBS and fixed with 4% paraformaldehyde for 15 min.

Proximity ligation assays (PLA) were performed using DuoLink In Situ Detection Reagents Orange (Sigma) according to manufacturer's recommendations using primary mouse anti-CD1d (K253, Biolegend, 1:100) and rabbit anti-CD36 antibodies (EPR6573, Abcam, 1:100).

Images were acquired using an inverted Zeiss LSM 710 microscope. At least 6 different images per experiment were analysed within Fiji (ImageJ).

## RNA sequencing
RNA was extracted from pMacs isolated from 4 WT and 4 CD1d-KO mice using the RNAeasy micro kit (Qiagen) following the manufacturer's instructions. RNA sequencing was performed as described[56]. Briefly, libraries were created using a polyA KAPA mRNA HyperPrep kit (Roche). Libraries were barcoded and run on an Illumina HiSeq

2500 system generating 25 million single end 75 bp reads per sample. Fastq files were trimmed using Cutadapt (version 1.9.1) with a quality threshold of 10 before being aligned to and quantified against ensemble GRCm38 release 86 of the mouse genome with RSEM (version 1.3.0)/STAR(version 2.5.2a).

The raw counts were then imported into R/Bioconductor (version 3.6.0). DESeq2 (version 1.24.0) was used to estimate different size factors between samples, and a model (covariates being genotype and replicate id) was used to find genes that were differentially expressed (Fold-change > 1.5, false discovery threshold of <0.01) between genotypes. Gene ontology analysis were performed using PANTHER analysis tools[57,58].

## Statistical analyses
Statistical analyses were performed using Prism software (GraphPad). Unless specified otherwise, *n* represents the number of individual mice analysed in each experiment. Statistical significance was determined using paired or unpaired two-tailed student's *t*-test, one-sample *t*-test, 1 or 2-way ANOVA or Mann-Whitney test as specified in the figure legends.

## Reporting summary
Further information on research design is available in the Nature Portfolio Reporting Summary linked to this article.

## Data availability
Sequencing data associated with this paper has been submitted to NCBI's GEO repository under the accession number GSE215837. Source data are provided with this paper.

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

## Acknowledgements

This work was funded by the UK Biotechnology and Biological Sciences Research Council (grants to P.B. BB/T013710/1 and BB/S005560/1); and the Cancer Research UK King's Health Partners Centre at King's College London (C604/A25135). P.M.B. was supported by a studentship from the UK Medical Research Council and King's College London Doctoral Training Partnership in Biomedical Sciences (MR/N013700/1). P.G. was supported by a Ser Cymru fellowship programme from the Welsh Government and EU-ERDF funds. The authors acknowledge technical support from the Biological Research Facility, the Advanced Sequencing, the Flow Cytometry and the Light Microscopy Platforms from The Francis Crick Institute (which receives its core funding from Cancer Research UK, the UK Medical Research Council and the Wellcome Trust).

## Author contributions

P.B. conceptualized and supervised the study; S.J., P.G. and V.O. contributed to experimental interpretation, discussion and editorial comments; P.M.B., L.E. and J.C.L.-R. conducted experiments and designed parts of the study; A.S. assisted with microscopy acquisition and analyses; G.K. performed RNAseq analyses; V.T. performed lipid extraction and lipidomics analyses under supervision of V.O.; P.B. wrote the paper which was revised and edited by all authors.

## Competing interests

The authors declare no competing interests.
