## [Peer Review File · Nature Communications]

CD1d-dependent rewiring of lipid metabolism in macrophages regulates innate immune responsesREVIEWER COMMENTS

Reviewer #1 (Remarks to the Author):

Brailey et al. Review

This manuscript by Brailey et al. describes a novel role for the CD1d molecule in regulating lipid metabolism in macrophages (and to some extent in dendritic cells as well). Cell-intrinsic CD1d deficiency leads to increased pro-inflammatory responses to innate stimuli (TLRs). This is accompanied by changes in lipid metabolism, mainly lipid uptake mediated by CD36.

This is an interesting finding and the data provided is convincing. The manuscript is brief and well written.

One of the main questions is whether the CD1d-mediated changes in lipid metabolism in macrophages (and/or DCs) are relevant in vivo in an infectious and/or inflammatory setting. Perhaps monocyte/macrophage-specific deletion of CD1d (cre or cre-ERT2 models) using CD1d^{fl/fl} mice in an obesity or atherosclerosis model, as suggested in the discussion.

The model in Sup Fig 4 needs to be substantiated. For example, the link between CD1d/CD36 and PPAR δ is solely based on expression of PPAR δ in CD1d-deficient macrophages. Further direct characterization is needed.

What is the basis for CD1d-mediated regulation of CD36 function? At least 2 different amino-acid (SAYQDIR and RRR) motifs have been identified in the cytoplasmic domain of CD1d and shown to regulate the intracellular trafficking of CD1d. Are these motifs, or other portions of the molecule necessary to regulate CD36 function? This could be done for example by reconstitution of CD1d-deficient macrophages. There might also be a mouse strain for the SAYQDIR motif (CD1d-TD, tail deleted).

Additional comments:

It should be made very clear which experiments were conducted with littermate mice, and which were not.

In Fig 1, treatment with polyI:C or E. coli do not reach statistical significance but is mentioned in the text as having the same effect as LPS and CpG.

The pie chart in Fig 3A is not very informative. Why not using a volcano plot and highlight some of the DEGs?

Some statistics are missing from Figures 2, Sup Fig 2 does not include statistics, making the data difficult to interpret.

I believe that tumor cells tend to downregulate or lose CD1d expression, which is viewed as an mechanism to escape immune pressure by iNKT cells. Could this be discussed in the introduction or conclusion. What could be the relevance of the proposed findings in that context?

Reviewer #2 (Remarks to the Author):

The manuscript by Brailey et al. has explored the effects of CD1d expression on macrophage function. They find that macrophages from CD1d-deficient mice respond more strongly to TLR signaling and exhibit metabolic alterations and increased lipid transport. Mechanistic studies revealed that CD1d influences CD36 internalization. From these findings the authors conclude that CD1d is a novel regulator of the inflammatory and metabolic circuits of macrophages, independently of its capacity to serve as a ligand for NKT cells.

General comments:

Although previous studies have hinted at alterations of CD1d-deficient macrophages to inflammatory stimuli, mechanisms have remained unknown. The studies reported here provide evidence for a mechanism involving CD36. The findings are convincing and the paper is well-written. The work is relevant to prior studies reporting various alterations in experimental disease models in CD1d-deficient mice, some of which did not always reproduce in Ja18-deficient mice (which still express CD1d but like CD1d-deficient mice lack iNKT cells).

Minor comments:

1. Although the authors note in the discussion that others have reported effects of CD1d-deficiency in macrophages on responses to inflammatory stimuli, it would be more appropriate to refer to these studies in the first section of the results.
2. Please consider noting in the discussion that CD36 and CD1d are expressed in a variety of cell types other than macrophages (e.g., adipocytes, enterocytes), where similar alterations as those reported in the paper might occur.
3. Please consider noting in the discussion that the findings might be relevant to CD1a and CD1b, which also bind CD36 family members.

Reviewer #3 (Remarks to the Author):

GENERAL:

The demonstration that CD1d has likely roles in regulating TLR responses via control of CD36 is novel and has implications for understanding macrophage inflammatory pathways and inflammatory disease processes. The authors show that CD1d-deficient macrophages (and BMDC) have enhanced TLR-inducible inflammatory responses and that lipid metabolism is dysregulated in these cells. Bone marrow chimera studies also confirm cell-intrinsic effects. Some evidence connects CD36 to the phenotypes that were identified. If the mechanistic links between CD1d, CD36 and inflammatory responses can be more convincingly demonstrated by the authors, this study would be of widespread interest to the innate immunity field. I do have some additional concerns about data presentation and interpretation, as highlighted below.

MAJOR:

1. In my view, there is some over-reliance on using CD1d knock-out macrophages to ascribe specific phenotypes to CD1d. This concern is somewhat alleviated by the fact that the authors used an independent CD1d knock-out line to confirm the hyperinflammatory

phenotype (Supplementary Fig 2F). Nonetheless, alternate/complementary approaches would strengthen the study considerably. For example, the authors show that a blocking antibody against CD1d (19G11) increased CD36 internalization in Fig 6G. Does this antibody also increase cytokine responses in wild type macrophages, but not CD1d-deficient macrophages, as per phenotypes in Fig 1? Or downregulate genes associated with lipid metabolism, as per phenotypes in Fig 3? An independent approach to validate key CD1d-linked phenotypes is highly desirable.

2. Are there TLR-inducible inflammatory responses that are not affected by CD1d deficiency? Or inflammatory responses downstream of other pro-inflammatory stimuli? Evidence for specificity in the inflammatory phenotypes would strengthen the conclusions that are drawn and provide reassurance that the phenotypes don't just reflect a general phenomenon relating to the state of the CD1d KO cells.

3. The authors show quite a selective effect of CD1d deficiency on Ppard mRNA in Fig 3, but this is not followed up on further. For example, is this also apparent at the protein level and does ectopic expression of Ppard in CD1d-deficient macrophages overcome defects in lipid metabolism and/or dysregulated CD36 internalisation and cytokine responses?

4. Data presentation could be improved considerably. Firstly, the manuscript suffers from over-normalising data throughout the manuscript, rather than presenting at least some key findings as raw data combined from independent experiments - ideally, it would be preferable to do so, if possible (though I appreciate that this can be difficult when absolute values may vary between experiments). At the very least, the authors could present individual data points from each experiment in their combined data for bar graphs that are presented (as they have done for the middle panels in Fig 1B, Fig 1C and Fig 1D). Secondly, the authors should not do two rounds of normalisation within the one data set – it looks like this has been done in Fig 6F? If so, the control KO values should not be normalised since one needs to see the baseline CD36 levels in CD1d knock-out cells versus wild type cells to be able to interpret these data appropriately. Finally, several data sets do not include unstimulated controls (e.g. Fig 1B middle and right, 1C, 1D middle and right, 5C, 5D, 5E). These should be included on the same graphs, if possible.

5. Since some of the hyperinflammatory phenotypes that are reported are reasonably modest in effect size (e.g. 1.5 fold increases), the authors consider further substantiating phenotypes with some time course or dose response data for cytokine production. I realise that some dose response data are included in Fig 1B and 1D - but only two LPS concentrations were used, and only representative experiments are shown.

6. Some of the data shown in Figure 2B and 2C do not look to be particularly compelling. If the authors wish to make claims about enhanced TLR signalling in CD1d KO BMDCs, it would be preferable to show quantified data combined from multiple independent experiments (similar to Fig 2A, right panel).

7. The Discussion could provide better coverage of existing literature on the role of CD36 in TLR signalling, along with potential explanations for the authors findings that link dysregulated CD36 function to enhanced inflammatory responses. The Discussion could also provide better coverage of other aspects of this study – for example, an explanation of how Ppard dysregulation might be predicted to impact on TLR responses.

MINOR:

1. Introduction, page 4, last paragraph: "...we uncover a new function for CD1d...". It is not a "new" function, so this text should be reworded e.g. to "previously unrecognized".
2. Results, page 5, first paragraph: second last line should refer to Supplementary Figure 1C (not 1B).
3. While I can understand why the authors have not included error bars for representative experiments (e.g. Fig 1B left, Fig 1D second panel), they could still show mean \pm range for technical replicates and indicate as such within legend – not for the purpose of statistical analysis (which would be inappropriate), but so that one could see the variability in the assay to better understand how robust the data are. Or alternatively, since some of these are important data sets, the authors could show combined raw data for multiple independent experiments to strengthen the data.
4. Some details are lacking in the methods. For example, there is no description of the specific LPS and CpG that are used in these studies. This is important for others to reproduce this work, as there are many different varieties of each of these TLR ligands (e.g. source, purity, stimulatory capacity).
5. Supplementary Fig 3 – in the spreadsheet showing differentially expressed genes, several entries do not have a gene symbol. Is this table correct or are some data missing?

Re: “CD1d-dependent rewiring of lipid metabolism in macrophages regulates responses to innate signals” by PM Brailey, L Evans, JC López-Rodríguez, A Sinadinos, V Tyrrel, G Kelly, V O’Donnell, P Ghazal, S John & P Barral.

Reply to reviewers

We are very grateful for the positive and constructive comments provided by the reviewers and their thoughtful questions. We believe that the additional experiments performed have strengthened the manuscript and provide further support to a previously unrecognised role for CD1d in the regulation of the immune-metabolic program in macrophages.

Reviewer #1 (R1)

This manuscript by Brailey et al. describes a novel role for the CD1d molecule in regulating lipid metabolism in macrophages (and to some extent in dendritic cells as well). Cell-intrinsic CD1d deficiency leads to increased pro-inflammatory responses to innate stimuli (TLRs). This is accompanied by changes in lipid metabolism, mainly lipid uptake mediated by CD36.

This is an interesting finding and the data provided is convincing. The manuscript is brief and well written.

1) One of the main questions is whether the CD1d-mediated changes in lipid metabolism in macrophages (and/or DCs) are relevant in vivo in an infectious and/or inflammatory setting. Perhaps monocyte/macrophage-specific deletion of CD1d (cre or cre-ERT2 models) using CD1dfl/fl mice in an obesity or atherosclerosis model, as suggested in the discussion.

We thank R1 for suggesting these additional experiments as they make a valuable contribution to our revised manuscript.

We have tested the relevance of the CD1d-dependent immune-metabolic axis in 2 settings: a model of LPS-induced sepsis and a model of bacterial peritonitis (**Fig 5E-H**). To examine the role of CD1d in these models, we depleted mice of endogenous macrophages and reconstituted them by adoptively transferring WT or CD1d-KO cells. To determine the relevance of lipid metabolism we blocked CD36-mediated lipid uptake *in vivo*, by injection of the CD36 irreversible inhibitor SSO.

In LPS-induced sepsis we found a CD1d-dependent increase in cytokine secretion as well as hypothermia both of which were restored in response to manipulation of metabolic pathways by blocking CD36. This new data is included in **Fig 5E-F**.

As a model of bacterial peritonitis, we selected intraperitoneal injection of *E. coli*, as this represents a clinically relevant infection with high mortality rate. In this model, CD1d-deficiency also lead to increased pro-inflammatory cytokine concentration in the blood, yet this resulted in restricted bacterial dissemination to peripheral organs, as we recovered less bacterial CFU from the lungs of mice receiving CD1d-KO cells. *In vivo* blocking of CD36 reverted this protective phenotype. This new data is included in **Fig 5G-H**.

Thus, these experiments further support and reinforce the role of CD1d in the regulation of an immune-metabolic axis with implications for inflammatory and infectious responses *in vivo*.

CD1dfl/fl mice crossed with LysM-Cre or Csf1r-Cre were not available to perform these experiments and generating these strains will unduly delay this manuscript. However, the adoptive transfer strategy into macrophage-depleted mice that we have employed has been widely reported (e.g. Kozicky & Sly, *Methods in Molecular Biology*, 2019; Jung et al, *Nat Comms* 2018; Di et al, *Immunity* 2018; Lui et al, *Nat Immunol* 2011; Du et al, *Dev Cell* 2020; Wang et al, *Sci Adv*, 2021; Xu et al, *Nat Immunol* 2012; Wang et al, *Front Immunol* 2022) and provides proof-of-concept evidence for the function of CD1d in controlling an immune-metabolic axis *in vivo*.

2) The model in Sup Fig 4 needs to be substantiated. For example, the link between CD1d/CD36 and PPARd is solely based on expression of PPARd in CD1d-deficient macrophages. Further direct characterization is needed.

We agree with the reviewer that the *Ppard* phenotype is striking, and we have performed further experiments to define its function in our system. PPAR β/δ is a lipid-activated transcription factor which contributes to the regulation of lipid homeostasis and has been proposed to play an anti-inflammatory role in immunity. Thus, we hypothesized that activation of PPAR δ may restore the hyper-activation of CD1d-KO cells.

To investigate the function of PPAR δ we first performed western-blot analyses and detected that PPAR δ (protein) levels are also reduced in CD1d-KO cells vs WT-in line with reduced mRNA levels (new **Sup Fig 4B**). Moreover, we further confirmed that *Ppard* expression levels are regulated downstream of CD1d functions as we found that blocking CD1d on the surface of WT cells (with a blocking antibody) was sufficient to reduce the expression of *Ppard* (new **Fig 6I**). Similar results were obtained when blocking CD1d in RAW246.7 macrophages (new **Sup Fig 5C**).

Next, we tested whether PPAR δ activation was sufficient to overcome the immune-metabolic reprogramming of CD1d-KO cells. To do this, we took advantage of a PPAR δ agonist (L165,041) which has been previously used to investigate the role of PPAR δ activation in macrophage biology (*Adhikary et al, Nucleic Acids Res, 2015*). Strikingly, activation of PPAR δ did restore lipid uptake, expression of metabolic genes and TLR-dependent responses in CD1d-KO cells to comparable levels to those of WT cells. This new data is included in **Fig 5I-K**

All together this new data further supports a role for PPAR δ as a mechanistic link between metabolic and immune responses in macrophages.

3) What is the basis for CD1d-mediated regulation of CD36 function? At least 2 different amino-acid (SAYQDIR and RRR) motifs have been identified in the cytoplasmic domain of CD1d and shown to regulate the intracellular trafficking of CD1d. Are these motifs, or other portions of the molecule necessary to regulate CD36 function? This could be done for example by reconstitution of CD1d-deficient macrophages. There might also be a mouse strain for the SAYQDIR motif (CD1d-TD, tail deleted).

To define the molecular requirements for the CD1d-dependent regulation of CD36 we compared the function of (WT) CD1d with 2 mutant forms: a tail deleted CD1d (CD1d-TD) lacking the SAYQDIR motif of the intracellular tail, and a point mutant CD1d Y \rightarrow A for Y332 (CD1d-Y). Both of these CD1d mutants have been previously shown to be expressed on the cell surface, yet they display abnormalities in intracellular trafficking and internalization (*Chiu et al, Nat Immunol 2001; Lawton et al, J Immunol 2005*).

To investigate the CD1d-dependent control of CD36-mediated lipid uptake we measured uptake of fluorescent acLDL in HEK293T cells expressing CD1d (WT, TD or Y) and/or CD36. In this system, lipid uptake is primarily dependent on CD36 expression as acLDL internalization is negligible in its absence. Expression of (WT) CD1d decreased CD36-mediated lipid uptake, and this decrease in uptake can be reversed by an α CD1d-blocking antibody. Strikingly, the regulation of lipid uptake by CD1d is independent on its intracellular tail, as both mutant CD1d forms show a comparable effect to their WT counterpart on acLDL import which can again be reverted by α CD1d. This new data is included in new **Fig 6K and Sup Fig 5E**

Thus, our new data indicate that the CD1d modulation of lipid uptake through CD36 is independent on the CD1d intracellular tail. It is also worth noting that lipid internalization can be increased by incubation of cells with a α CD1d antibody (Fig 6H, Fig 6K, Sup Fig 5B), suggesting that surface CD1d -rather than the intracellular portion- is important for this function.

Additional comments:

It should be made very clear which experiments were conducted with littermate mice, and which were not.

For our experiments we have used either littermates or age and sex matched WT mice (C57Bl/6) as controls. The majority of the stimulation experiments (Fig 1) as well as RNA

sequencing experiments (Fig 3) were done with CD1d-KO and littermate controls. We have later repeated the experiments using age and sex matched WT mice as controls and got identical results (we have included this comparison in **Sup Fig 2G**). We have specified throughout the manuscript when littermate controls have been used.

In Fig 1, treatment with polyI:C or E. coli do not reach statistical significance but is mentioned in the text as having the same effect as LPS and CpG.

We apologize for the confusion. It is true that stimulation with poly I:C and *E. coli* do not reach significance (WT vs KO) in terms of IFN- β secretion, yet they do for IL-6 or TNF- α secretion (Sup Fig 2). We have clarified this in the text (**Page 6**).

The pie chart in Fig 3A is not very informative. Why not using a volcano plot and highlight some of the DEGs?

We have included a volcano plot in **Fig 3A**

Some statistics are missing from Figures 2, Sup Fig 2 does not include statistics, making the data difficult to interpret.

We have included additional quantifications and statistics in **Fig 2** and **Sup Fig 2**.

I believe that tumor cells tend to downregulate or lose CD1d expression, which is viewed as an mechanism to escape immune pressure by iNKT cells. Could this be discussed in the introduction or conclusion. What could be the relevance of the proposed findings in that context?

We thank the reviewer for pointing this out, this is indeed an interesting point and we have included it in the discussion (**Page 19**).

Reviewer #2 (R2)

The manuscript by Brailey et al. has explored the effects of CD1d expression on macrophage function. They find that macrophages from CD1d-deficient mice respond more strongly to TLR signaling and exhibit metabolic alterations and increased lipid transport. Mechanistic studies revealed that CD1d influences CD36 internalization. From these findings the authors conclude that CD1d is a novel regulator of the inflammatory and metabolic circuits of macrophages, independently of its capacity to serve as a ligand for NKT cells.

General comments:

Although previous studies have hinted at alterations of CD1d-deficient macrophages to inflammatory stimuli, mechanisms have remained unknown. The studies reported here provide evidence for a mechanism involving CD36. The findings are convincing and the paper is well-written. The work is relevant to prior studies reporting various alterations in experimental disease models in CD1d-deficient mice, some of which did not always reproduce in Ja18-deficient mice (which still express CD1d but like CD1d-deficient mice lack iNKT cells).

We thank the reviewer for his/her appreciation of the relevance and novelty of our work

Minor comments:

1. Although the authors note in the discussion that others have reported effects of CD1d-deficiency in macrophages on responses to inflammatory stimuli, it would be more appropriate to refer to these studies in the first section of the results.

We agree with this suggestion and have referenced previous studies at the beginning of the results section (**Page 5**).

2. Please consider noting in the discussion that CD36 and CD1d are expressed in a variety of cell types other than macrophages (e.g., adipocytes, enterocytes), where similar alterations as those reported in the paper might occur.

We thank R2 for this suggestion and we have included this point in the discussion (**Page 18-19**)

3. Please consider noting in the discussion that the findings might be relevant to CD1a and CD1b, which also bind CD36 family members.

We agree this is an interesting point and have included it in the discussion (**Page 18**)

Reviewer #3 (R3)

GENERAL:

The demonstration that CD1d has likely roles in regulating TLR responses via control of CD36 is novel and has implications for understanding macrophage inflammatory pathways and inflammatory disease processes. The authors show that CD1d-deficient macrophages (and BMDC) have enhanced TLR-inducible inflammatory responses and that lipid metabolism is dysregulated in these cells. Bone marrow chimera studies also confirm cell-intrinsic effects. Some evidence connects CD36 to the phenotypes that were identified. If the mechanistic links between CD1d, CD36 and inflammatory responses can be more convincingly demonstrated by the authors, this study would be of widespread interest to the innate immunity field. I do have some additional concerns about data presentation and interpretation, as highlighted below.

MAJOR:

1. In my view, there is some over-reliance on using CD1d knock-out macrophages to ascribe specific phenotypes to CD1d. This concern is somewhat alleviated by the fact that the authors used an independent CD1d knock-out line to confirm the hyperinflammatory phenotype (Supplementary Fig 2F). Nonetheless, alternate/complementary approaches would strengthen the study considerably. For example, the authors show that a blocking antibody against CD1d (19G11) increased CD36 internalization in Fig 6G. Does this antibody also increase cytokine responses in wild type macrophages, but not CD1d-deficient macrophages, as per phenotypes in Fig 1? Or downregulate genes associated with lipid metabolism, as per phenotypes in Fig 3? An independent approach to validate key CD1d-linked phenotypes is highly desirable.

We agree with the reviewer that this is an important question and we have performed experiments that further validate the functions of CD1d using different approaches and cell types.

First, we measured the effect of CD1d-blocking in primary cells. Importantly, incubation of macrophages with a blocking antibody against CD1d not only increases CD36 internalization and lipid uptake, but it is sufficient to induce a downregulation of lipid metabolic genes as well as *Ppard*, and resulted in an increase in cytokine/ISG expression in response to TLR stimulation. Thus, incubation of WT macrophages with a CD1d blocking antibody recapitulates the phenotype of CD1d-KO cells. This new data is included in **Fig 6I-J**

To complement the above experiments, we have performed further essays with RAW264.7 macrophages (which express high levels of endogenous CD1d and CD36). Again, blocking CD1d with a blocking antibody was sufficient to increase lipid uptake in these cells, downregulate lipid metabolic genes (and *Ppard*) and increase responses to TLR stimulation. This new data is included in **Sup Fig 5A-D**.

Finally, we set up a system to define the molecular requirement for the CD1d-dependent regulation of CD36 function. To do this, we ectopically expressed CD1d and/or CD36 (as well as various CD1d mutants) in HEK293T cells and found that CD1d controls CD36-mediated lipid uptake in this system through a mechanism independent on the CD1d intracellular tail. This new data is included in **Fig 6K and Sup Fig 5E**

Thus, collectively these experiments further reinforce the function of CD1d in modulating CD36-mediated lipid uptake and the subsequent regulation of metabolic and immune responses

2. Are there TLR-inducible inflammatory responses that are not affected by CD1d deficiency? Or inflammatory responses downstream of other pro-inflammatory stimuli? Evidence for specificity in the inflammatory phenotypes would strengthen the conclusions that are drawn and provide reassurance that the phenotypes don't just reflect a general phenomenon relating to the state of the CD1d KO cells.

We have performed a number of additional stimulations and measured a variety of cytokines/ISGs to further explore the inflammatory phenotype of CD1d-KO cells.

To get an overview of the cytokine profile secreted by WT and CD1d-KO cells in response to TLR stimulation we took advantage of a cytometric bead array to quantify 13 different cytokines/chemokines in the culture supernatant (IL-6, TNF- α , MCP-1, IL-1 α , IL-1 β , IL-10, IL-12p70, IL-17A, IL-23, IL-27, GM-CSF, IFN- γ , IFN- β).

In response to TLR stimulation pMacs secreted IL-6, TNF- α and IFN- β , but also detectable levels of IL-10, yet IL-10 production was comparable for WT and KO cells (all other cytokines were undetectable). This data is included in **Fig 1B**.

We performed comparable experiments with BMDCs. In this case in addition to IL-6, TNF- α and IFN- β , BMDCs also secreted detectable amounts of MCP-1, with higher secretion detected for CD1d-KO BMDCs. This new data is included in **Sup Fig 2E**.

To further explore the inflammatory response of WT/CD1d-KO cells we stimulated BMDCs with a variety of cytokines: IFN- β (measuring ISG expression as a read-out), TNF α (measuring *Il1b*), IFN- γ (measuring *Tnfa*). In all these conditions responses were comparable for WT and CD1d-KO cells. This new data is included in **Sup Fig 2H**.

3. The authors show quite a selective effect of CD1d deficiency on Ppard mRNA in Fig 3, but this is not followed up on further. For example, is this also apparent at the protein level and does ectopic expression of Ppard in CD1d-deficient macrophages overcome defects in lipid metabolism and/or dysregulated CD36 internalisation and cytokine responses?

We agree with the reviewer that the *Ppard* phenotype is striking, and we have performed further experiments to define its function in our system. PPAR β/δ is a lipid-activated transcription factor which contributes to the regulation of lipid homeostasis and has been proposed to play an anti-inflammatory role in immunity. Thus, we hypothesized that activation of PPAR δ may restore the hyper-activation of CD1d-KO cells.

To investigate the function of PPAR δ we first performed western-blot analyses and detected that PPAR δ (protein) levels are also reduced in CD1d-KO cells vs WT-in line with reduced mRNA levels (new **Sup Fig 4B**). Moreover, we further confirmed that *Ppard* expression levels are regulated downstream of CD1d functions as we found that blocking CD1d on the surface of WT cells (with a blocking antibody) was sufficient to reduce the expression of *Ppard* (new **Fig 6I**). Similar results were obtained when blocking CD1d in RAW246.7 macrophages (new **Sup Fig 5C**).

Next, we tested whether PPAR δ activation was sufficient to overcome the immune-metabolic reprogramming of CD1d-KO cells. To do this, we took advantage of a PPAR δ agonist (L165,041) which has been previously used to investigate the role of PPAR δ activation in macrophage biology (*Adhikary et al, Nucleic Acids Res, 2015*). Strikingly, activation of PPAR δ did restore lipid uptake, expression of metabolic genes and TLR-dependent responses in CD1d-KO cells to comparable levels to those of WT cells. This new data is included in **Fig 5I-K**

All together this new data further supports a role for PPAR δ as a mechanistic link between metabolic and immune responses in macrophages.

4. Data presentation could be improved considerably. Firstly, the manuscript suffers from over-normalising data throughout the manuscript, rather than presenting at least some key findings as raw data combined from independent experiments - ideally, it would be preferable to do so, if possible (though I appreciate that this can be difficult when absolute values may vary between experiments). At the very least, the authors could present individual data points from each experiment in their combined data for bar graphs that are presented (as they have done for the middle panels in Fig 1B, Fig 1C and Fig 1D). Secondly, the authors should not do two rounds of normalisation within the one data set – it looks like this has been done in Fig 6F? If so, the control KO values should not be normalised since one needs to see the baseline CD36 levels in CD1d knock-out cells versus wild type cells to be able to interpret these data appropriately. Finally, several data sets do not include unstimulated controls (e.g. Fig 1B middle and right, 1C, 1D middle and right, 5C, 5D, 5E). These should be

included on the same graphs, if possible.

Indeed, we originally didn't pool experimental data because absolute values differ from experiment to experiment (yet the net change is consistent across experiments). We have revisited our data, repeated some experiments and pooled data from individual experiments whenever possible (instead of showing representative ones) and show individual data points in **all figures** throughout the paper.

In Fig 6F there is only one round of normalization in which each sample is normalised to its own control (WT+oxLDL to WT; KO+oxLDL to KO). This enables direct comparison of the degree of CD36 internalization between WT and KO cells in response to oxLDL. For clarity, we have included the raw MFI data for CD36 in individual WT and KO samples before and after oxLDL treatment in a new panel in **Fig 6F**.

We have included unstimulated controls in Figures: **Fig 1B, Fig 1C, Fig 1D, Sup Fig 1C, Sup Figs 2B, 2D, 2E, 2H**. For Fig 5 unstimulated controls for cytokine secretion (+/-SSO) are included in **Sup Fig 4A**. For the qPCRs in Fig 5 we are unable to provide data for non-stimulated controls as values are shown relative to stimulated WT cells. However, we did perform these controls and found that ISG/cytokine expression levels are unchanged by SSO or α CD36 in the absence of TLR stimulation.

5. Since some of the hyperinflammatory phenotypes that are reported are reasonably modest in effect size (e.g. 1.5 fold increases), the authors consider further substantiating phenotypes with some time course or dose response data for cytokine production. I realise that some dose response data are included in Fig 1B and 1D - but only two LPS concentrations were used, and only representative experiments are shown.

We have added further titration experiments in **Fig 1B** and **Sup Fig1C** and show pooled data from independent experiments across the manuscript as detailed above

6. Some of the data shown in Figure 2B and 2C do not look to be particularly compelling. If the authors wish to make claims about enhanced TLR signalling in CD1d KO BMDCs, it would be preferable to show quantified data combined from multiple independent experiments (similar to Fig 2A, right panel).

We have included quantifications for independent experiments in **Figs 2B** and **2C**

7. The Discussion could provide better coverage of existing literature on the role of CD36 in TLR signalling, along with potential explanations for the authors findings that link dysregulated CD36 function to enhanced inflammatory responses. The Discussion could also provide better coverage of other aspects of this study – for example, an explanation of how Ppard dysregulation might be predicted to impact on TLR responses.

We thank R3 for these suggestions. We have extended the discussion to cover the suggested aspects (**Pages 18-20**)

MINOR:

1. Introduction, page 4, last paragraph: "...we uncover a new function for CD1d....". It is not a "new" function, so this text should be reworded e.g. to "previously unrecognized".

We agree with this point and the text has been amended accordingly

2. Results, page 5, first paragraph: second last line should refer to Supplementary Figure 1C (not 1B).

We thank R3 for pointing this out.

3. While I can understand why the authors have not included error bars for representative experiments (e.g. Fig 1B left, Fig 1D second panel), they could still show mean±range for technical replicates and indicate as such within legend – not for the purpose of statistical analysis (which would be inappropriate), but so that one could see the variability in the assay to better understand how robust the data are. Or alternatively, since some of these are important data sets, the authors could show combined raw data for multiple independent experiments to strengthen the data.

We thank the reviewer for this suggestion. We have revisited our datasets and we now show combined raw data from several independent experiments as detailed above.

4. Some details are lacking in the methods. For example, there is no description of the specific LPS and CpG that are used in these studies. This is important for others to reproduce this work, as there are many different varieties of each of these TLR ligands (e.g. source, purity, stimulatory capacity).

We apologize for the omission. This information has been included in the methods (**Page 23**).

5. Supplementary Fig 3 – in the spreadsheet showing differentially expressed genes, several entries do not have a gene symbol. Is this table correct or are some data missing?

The table is actually correct, but the genes without symbols include pseudogenes, long non-coding RNAs, anti-sense transcripts or genes without functional annotation or a canonical name as predicted by ENSEMBL. When doing gene ontology analyses (in Panther) these genes are not mapped to a corresponding protein record. We have clarified this in the manuscript (**Page 8**).

REVIEWERS' COMMENTS

Reviewer #1 (Remarks to the Author):

The authors have addressed my comments and questions appropriately in their revised manuscript. They have provided substantial additional data that confirm their initial findings and underscore the in vivo biological relevance of CD1d-mediated regulation of lipid uptake. As such, this revised manuscript is stronger than its original version.

Reviewer #2 (Remarks to the Author):

The authors have responded appropriately to the reviewer comments.

Reviewer #3 (Remarks to the Author):

GENERAL:

The revised manuscript has been substantially improved through the inclusion of textual modifications, as well as several new data sets. These include data showing that CD36 antagonism attenuates the hyperinflammatory phenotype of CD1d deficiency in vivo; that PPAR-delta activation restores metabolic and inflammatory processes in CD1d-deficient macrophages; that an anti-CD1d antibody phenocopies the effects of CD1d deficiency; and that ectopic expression of CD1d in HEK cells attenuates CD36-mediated lipid uptake. With these additions and other modifications, the authors have thoroughly addressed the various issues that I had raised in my original review. In my view, the findings in this manuscript will be of great interest to those working in innate immunity and/or immunometabolism. I do have a few very minor comments on the revised manuscript for the authors to consider.

MINOR COMMENTS:

1. 2nd page of results (pg 6, lower half of page): The authors don't appear to describe the data in Supplementary Fig 2F showing that CD1d deficiency does not affect Tlr4 mRNA expression anywhere in the results section – these findings should be described at the appropriate point of the results text e.g. between description of Supplementary Fig 2E and Supplementary Fig 2G. Alternatively (and perhaps more appropriately), the figure panel could be presented at the end of Supplementary Fig 2 (e.g. after current Fig S2I), then described at the bottom of page 7/top of page 8 where the TLR4 internalisation data (Fig 2D) are described.
2. Legend to Figure 1B: It is stated that data are pooled from 2-9 experiments - statistical analysis should not be performed on n=2.
3. Individual data points have now been added to most of the figures, which is good to see. However, the choice of colour and background makes it difficult to visualise the data points in many of the figure panels. The authors could consider revising colour schemes for either bar graphs or the data points, such that the data points on bar graphs are clearly apparent (e.g. as is the case in the middle panels of Figure 1B, Figure 1C and Figure 1D – in these cases, all data points are very obvious).
4. For figure panels that show immunoblots, it is recommended that marker sizes on the

cropped blots are displayed, if possible. My apologies that I failed to note this in my comments on the original manuscript.

Re: “CD1d-dependent rewiring of lipid metabolism in macrophages regulates innate immune responses” by PM Brailey, L Evans, JC López-Rodríguez, A Sinadinos, V Tyrrel, G Kelly, V O'Donnell, P Ghazal, S John & P Barral.

Reply to reviewers

We are very grateful for the positive and constructive comments provided by the reviewers and their recognition of the relevance of our work.

Reviewer #1 (R1)

The authors have addressed my comments and questions appropriately in their revised manuscript. They have provided substantial additional data that confirm their initial findings and underscore the in vivo biological relevance of CD1d-mediated regulation of lipid uptake. As such, this revised manuscript is stronger than its original version.

We thank the reviewer for his/her appreciation of the work

Reviewer #2 (R2)

The authors have responded appropriately to the reviewer comments.

We thank the reviewer for his/her appreciation of the work

Reviewer #3 (R3)

GENERAL:

The revised manuscript has been substantially improved through the inclusion of textual modifications, as well as several new data sets. These include data showing that CD36 antagonism attenuates the hyperinflammatory phenotype of CD1d deficiency in vivo; that PPAR-delta activation restores metabolic and inflammatory processes in CD1d-deficient macrophages; that an anti-CD1d antibody phenocopies the effects of CD1d deficiency; and that ectopic expression of CD1d in HEK cells attenuates CD36-mediated lipid uptake. With these additions and other modifications, the authors have thoroughly addressed the various issues that I had raised in my original review. In my view, the findings in this manuscript will be of great interest to those working in innate immunity and/or immunometabolism. I do have a few very minor comments on the revised manuscript for the authors to consider.

We thank the reviewer for highlighting the novelty and relevance of our work

MINOR COMMENTS:

1. 2nd page of results (pg 6, lower half of page): The authors don't appear to describe the data in Supplementary Fig 2F showing that CD1d deficiency does not affect Tlr4 mRNA expression anywhere in the results section – these findings should be described at the appropriate point of the results text e.g. between description of Supplementary Fig 2E and Supplementary Fig 2G. Alternatively (and perhaps more appropriately), the figure panel could be presented at the end of Supplementary Fig 2 (e.g. after current Fig S2I), then described at the bottom of page 7/top of page 8 where the TLR4 internalisation data (Fig 2D) are described.

We thank the reviewer for noticing this omission. We have included the description of this panel in the results section (Page 6).

2. Legend to Figure 1B: It is stated that data are pooled from 2-9 experiments - statistical analysis should not be performed on n=2.

Apologies, this has been corrected

3. Individual data points have now been added to most of the figures, which is good to see. However, the choice of colour and background makes it difficult to visualise the data points in many of the figure panels. The authors could consider revising colour schemes for either bar graphs or the data points, such that the data points on bar graphs are clearly apparent (e.g. as is the case in the middle panels of Figure 1B, Figure 1C and Figure 1D – in these cases, all data points are very obvious).

We have modified all panels in the paper, so individual data points are clearly visible in each figure

4. For figure panels that show immunoblots, it is recommended that marker sizes on the cropped blots are displayed, if possible. My apologies that I failed to note this in my comments on the original manuscript.

Markers have been added to the western-blots